# Benzoxazole-derivatives enhance progranulin expression and reverse the aberrant lysosomal proteome caused by GRN haploinsufficiency

Rachel Tesla[1,2,10], Charlotte Guhl[3,10], Gordon C. Werthmann[1,2,10], Danielle Dixon[1,2], Basar Cenik[1,2], Yesu Addepalli[4], Jue Liang[4], Daniel M. Fass[5], Zachary Rosenthal[6,7], Stephen J. Haggarty[5], Noelle S. Williams[4], Bruce A. Posner[4], Joseph M. Ready[4] & Joachim Herz[1,2,8,9]✉

Heterozygous loss-of-function mutations in the *GRN* gene are a major cause of hereditary frontotemporal dementia. The mechanisms linking frontotemporal dementia pathogenesis to progranulin deficiency are not well understood, and there is currently no treatment. Our strategy to prevent the onset and progression of frontotemporal dementia in patients with *GRN* mutations is to utilize small molecule positive regulators of *GRN* expression to boost progranulin levels from the remaining functional *GRN* allele, thus restoring progranulin levels back to normal within the brain. This work describes a series of blood-brain-barrier-penetrant small molecules which significantly increase progranulin protein levels in human cellular models, correct progranulin protein deficiency in *Grn*[+/−] mouse brains, and reverse lysosomal proteome aberrations, a phenotypic hallmark of frontotemporal dementia, more efficiently than the previously described small molecule suberoylanilide hydroxamic acid. These molecules will allow further elucidation of the cellular functions of progranulin and its role in frontotemporal dementia and will also serve as lead structures for further drug development.

Frontotemporal dementia (FTD) is the second most common form of early-onset dementia and is currently incurable[1–3]. It is characterized by progressive, neuronal degeneration of the prefrontal and anterior temporal lobes and has a highly heterogeneous clinical representation with cognitive decline, language deficits and behavioral changes in personality and social conduct[4–7]. Currently the only treatment options for FTD are limited to pharmacological relief of disease symptoms[8,9]. No therapies are available to delay or stop neurodegeneration.

In 2006, heterozygous loss-of-function mutations within the progranulin (PGRN) encoding *GRN* gene were identified as a cause of

[1]Department of Molecular Genetics, University of Texas Southwestern Medical Center, Dallas, TX, USA. [2]Center for Translational Neurodegeneration Research, Dallas, TX, USA. [3]Faculty of Chemistry and Earth Sciences, Institute of Organic Chemistry and Macromolecular Chemistry, Friedrich Schiller University Jena, 07743 Jena, Germany. [4]Department of Biochemistry, University of Texas Southwestern Medical Center, Dallas, TX, USA. [5]Chemical Neurobiology Laboratory, Center for Genomic Medicine, Departments of Neurology and Psychiatry, Massachusetts General Hospital, Harvard Medical School, Boston, MA, USA. [6]Chemical Neurobiology Laboratory, Center for Genomic Medicine, Massachusetts General Hospital, Boston, MA, USA. [7]Department of Chemistry & Chemical Biology, Harvard University, Cambridge, MA, USA. [8]Department of Neuroscience, University of Texas Southwestern Medical Center, Dallas, TX, USA. [9]Department of Neurology and Neurotherapeutics, University of Texas Southwestern Medical Center, Dallas, TX, USA. [10]These authors contributed equally: Rachel Tesla, Charlotte Guhl, Gordon C. Werthmann. ✉e-mail: Joachim.Herz@utsouthwestern.edu

FTD, accounting for up to 20% of all genetic FTD cases[10–14]. The majority of mutations result in nonsense-mediated decay and PGRN haploinsufficiency[10,12,14–17]. Lipofuscin accumulation, dysregulated lysosomal lipid homeostasis, impaired autophagy, neuropathological aggregation of the RNA-binding protein TDP-43, and inflammation are key pathogenic hallmarks of FTD[18–25]. Further, homozygous loss of functional PGRN causes neuronal ceroid lipofuscinosis CLN11, an early-onset, neurodegenerative, lysosomal storage disorder[26–28]. Unfortunately the mechanisms involved in neurodegeneration associated with PGRN deficiency remain elusive.

One conjecture is that a key driver of PGRN deficiency related neurodegeneration is lysosomal dysfunction[29]. Impairments of autophagic flux[30], lysosomal cathepsin D deficiency[31], aberration of the lysosomal proteome[25,32] and lipidome[23] due to PGRN deficiency point to an important role of PGRN in lysosomal homeostasis.

PGRN is a highly conserved, secreted 88 kDa glycoprotein primarily expressed by microglia and neurons in the central nervous system (CNS)[33,34]. Full-length PGRN is credited with neuroprotective and anti-inflammatory properties[35–41] and its transcription is regulated alongside lysosomal biogenesis, autophagy, and cellular energy homeostasis genes[42–44].

When endocytosed, PGRN is transported to the lysosome where it is processed into seven granulin subunits[35,45–50]. The functions of individual granulins are a focus of contemporary research[51–54], but a role in lysosomal acidification[55] and regulation of lysosomal enzymatic activity[31,56–62] has been proposed. This underscores PGRN's pivotal role in maintaining lysosomal homeostasis with relevance for multiple CNS disorders[24,26,30,63–67].

Earlier work has used various methods to boost PGRN levels in FTD models[9,68–78]. These methods include adeno-associated virus (AAV) and recombinant protein-based therapies[79–82]. However, beyond preclinical animal studies, the delivery of viruses and proteins to all regions of the human brain presents significant challenges compared to administration of small molecules. Also, since patients likely need to be targeted and treated before the onset of symptoms and for the remainder of their lives, protein-based therapies could be cost-prohibitive and inefficient[83,84].

Previous work from our group demonstrated administration of the histone deacetylase (HDAC) inhibitor suberoylanilide hydroxamic acid (SAHA; vorinostat) increased the expression of progranulin protein levels in vitro[85]. However, due to the poor side effect profile of SAHA as well as the limited PGRN-boosting effects we observed in vivo, we moved forward to develop non-toxic and reliable enhancers of PGRN both in vitro and in vivo.

This study uses small molecules to take advantage of the remaining functional allele of the *GRN* gene in PGRN-dependent FTD patients for PGRN protein production to restore PGRN levels back to normal. To accomplish this objective, we conduct high-throughput screening (HTS) to identify small-molecule PGRN enhancers that do not alter protein maturation, transport etc., and develop analogs of selected hits through a structure-activity-relationship (SAR) approach. Both initial hits and analog compounds are tested for the ability to increase *Grn* mRNA and PGRN protein levels. Subsequently, promising compounds are evaluated in vitro and in vivo using pharmacokinetic and pharmacodynamic profiling. Here we describe small molecules with the ability to restore PGRN levels in PGRN haploinsufficient models in vitro and in vivo and the ability to correct aberrant lysosomal proteomes in FTD patient-derived cells. These small molecules will serve as lead structures for further development, optimization, and translational experiments.

## Results

### PGRN enhancers identified through high-throughput screening

Neuronally derived mouse Neuro-2a (N2A) cells stably transfected with a luciferase reporter under the control of the human *GRN* promoter were created to assess the effects of small-molecule treatment on *GRN* activation[85]. Approximately 200,000 small molecules were screened to probe for upregulation of *GRN* at the University of Texas Southwestern Medical Center (UTSW) HTS Core. Any small molecules with activity 3 standard deviations above the robust mean of the test population were considered a hit in the screen. Using this low-stringency cut-off, 15,820 compounds qualified as primary hits. Of these primary hits, 2144 compounds had a score of >100 and were marked for further evaluation (Supplementary Fig. 1). To further reduce our hits, we eliminated 87 small molecules which were considered to have generalized toxicity in twelve different non-small cell lung cancer (NSCLC) cell lines (Supplementary Fig. 1, red)[86]. We then chose the remaining top 1,280 hits to conduct a 3-point dose response activity confirmation experiment using the original *GRN* luciferase-based assay (Supplementary Fig. 2, blue).

### Unique compounds induce PGRN protein production in vitro

We next tested the top 127 commercially available compounds from the HTS small molecule confirmation experiment for their ability to upregulate PGRN. Each of the 127 selected compounds were tested for *GRN*-linked luciferase activation (Supplementary Fig. 2a) and toxicity (Supplementary Fig. 2b) over a dose range from 10 nM to 30 μM. The ATP-based CellTiterGlo™ (Promega) was utilized to assess toxicity. All compound doses resulting in more than a 20% reduction in ATP levels were excluded from further testing. After toxicity exclusions we selected the four most active concentrations of each compound to test their ability to induce PGRN protein expression. The treatment concentrations ranged from 100 nM-30 μM. N2A cells were harvested after 48 h of treatment and cell lysates were immunoblotted to evaluate compound activity. Nine compounds(C), C15, C19, C40, C41, C66, C105, C107, C116, and C127 (Fig. 1A) increased PGRN protein to approximately our target value of 2x compared to vehicle levels (Fig. 1B, C) and were selected for further evaluation.

### Three compounds boost PGRN levels in an in vivo model of FTD

After validating our selected compounds in vitro, we utilized an in vivo mouse model of FTD, a CNS disorder, to evaluate these compounds' activity in the brain. We used intracerebroventricular (ICV) cannulation with osmotic pumps to deliver the compounds directly to the brain to avoid complications due to differences in blood-brain-barrier (BBB) penetrance, distribution, and metabolic stability. All compounds were loaded into pumps at a concentration of 10 μM pre-delivery. Animals were infused at a delivery rate of 1 μL per hour and sacrificed after 7 days. Brains were collected for immunoblotting (Fig. 1D). Three compounds, C40, C41 and C127, produced a statistically significant increase of PGRN protein similar to WT ($Grn^{+/+}$) levels (Fig. 1E).

### Pharmacokinetic profiling

Having established that C40, C41, and C127 were active in vivo when delivered directly to the brain, we next evaluated their pharmacokinetic (PK) profiles when given peripherally. Several in vitro experiments were conducted to predict drug absorption, distribution, metabolism, and excretion (ADME) (Supplementary Fig. 3; Supplementary Table 1). First, compounds were incubated with murine microsome S9 fractions to simulate liver metabolism. C40 and C127 had satisfactory half-lives of 72 min and 76 min respectively, while C41 was eliminated from further PK testing due to its low half-life of 8 min (Supplementary Fig. 3a). Next, plasma stability of the compounds was assessed by incubation in murine plasma and saline. Both C40 and C127 exhibited acceptable half-lives of 533 min and 139 min, respectively (Supplementary Fig. 3b). Finally, in the intestinal permeability assay, both compounds exhibited no signs of active transport into the intestinal lumen when compared to controls, and both displayed adequate permeability (Supplementary Table 1). C40 and C127 PK profiles were further evaluated in vivo. Treatment concentrations of 10 mg/kg were used due to reduced

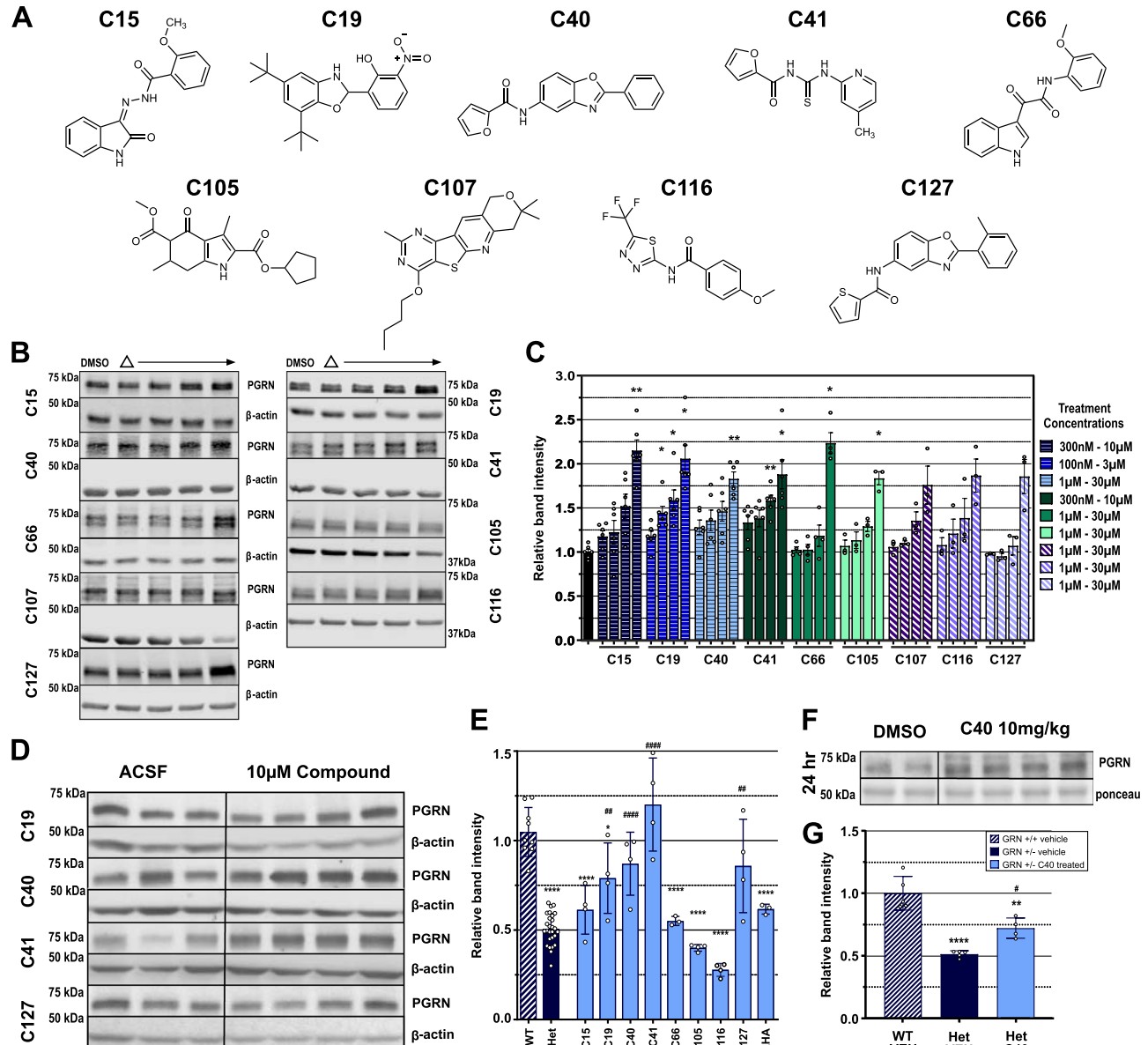

**Fig. 1 | Small molecules induced PGRN protein production in a dose dependent manner in vitro and in vivo. A** Molecule structure and chemical name of active compounds. **B** Immunoblots of active compounds with increasing doses(Δ). Compound abbreviation listed to the side of each respective blot. **C** Quantification of treatment replicates. Biological replicates of $n = 6$ for C15, C19, C40 and C41, $n = 4$ for C66 and $n = 3$ for C105, C107, C116 and C127 are represented as mean value ± SD and *$p < 0.05$, **$p < 0.01$, ***$p < 0.001$, ****$p < 0.0001$; All significance calculated using Brown-Forsythe and Welch ANOVA with Dunnett's T3 multiple comparisons test. **D, E** Seven days after pump implantation, $Grn^{+/-}$ mice treated with C40, C41 and C127 showed rescue of PGRN protein levels compared to $Grn^{+/+}$ mice. ($Grn^{+/+}$ $n = 4$, $Grn^{+/-}$ treated with ACSF $n = 17$, $Grn^{+/-}$ treated with drug n = 4/drug) **F, G** I.P. delivery of 10 mg/kg C40 significantly increased PGRN protein levels in $Grn^{+/-}$ mouse brains ($n = 4$) compared to vehicle treated $Grn^{+/-}$ mice ($n = 5$) after 24 h but did not return PGRN levels to $Grn^{+/+}$ levels. $Grn^{+/+}$ mice ($n = 5$) treated with vehicle serve as a control for healthy PGRN levels. For **D, G** All biological replicates are summarized in mean values ± SD and * represents $p < 0.05$, **$p < 0.01$, ***$p < 0.001$, ****$p < 0.0001$ with respect to $Grn^{+/+}$ ACSF; # represents $p < 0.05$, ##$p < 0.01$, ###$p < 0.001$, ####$p < 0.0001$ with respect to $Grn^{+/-}$ ACSF; ordinary one-way ANOVA with Tukey's multiple comparisons test (**E**) or Dunnett's multiple comparisons test (**G**) for significance was used.

solubility of both compounds at higher concentrations. C40 and C127 were found to be BBB-penetrant after intraperitoneal (i.p.) injection (Supplementary Fig. 3c, d). The concentration peaked at similar time points for both compounds, but the level of drug absorption differed significantly. C40 plasma and brain maximum observed concentrations ($C_{MAX}$) of 1498 ng/mL and 6246 ng/g, respectively, were significantly higher than that for C127 $C_{MAX}$ of 522 ng/mL and 1112 ng/g. However, both compounds demonstrated similar concentrations at 180 min. The $C_{MAX}$ was higher in the brain than in the plasma for both compounds, and the terminal half-life ($t_{1/2}$) of each was similar at 59 min for C127 and 41 min for C40. Next, mice were dosed with C40 using oral gavage to

compare per os (PO) vs. i.p. compound PK (Supplementary Fig. 3c, d). Oral $t_{1/2}$ was longer than i.p. $t_{1/2}$ at 65 min, and plasma and brain $C_{MAX}$ achieved 657 ng/mL and 1456 ng/g, respectively.

To evaluate enhancement of brain PGRN production in vivo, we treated $Grn^{+/-}$ mice i.p. with either 10 mg/kg C40 or vehicle. Whole mouse brains, which were harvested 24 h after injection and evaluated by western blotting, revealed that C40 significantly increases PGRN protein with respect to vehicle treated $Grn^{+/-}$ controls (Fig. 1F, G).

We reasoned that the efficacy of C40 and C127 might be limited by their ability to dissolve in solution in vivo (maximal dose of 10 mg/kg) and in vitro (maximal dose of 30 µM); therefore, we elected to produce

analog (A) compounds to improve their activity and physicochemical properties.

## Structure-Activity-Relationship (SAR) approach to increase solubility and activity

SAR mapping can be a valuable tool to improve lead compounds that do not yet possess optimal pharmacological properties. Thus, we conducted an approach to optimize activity and improve solubility relative to C40 and C127. We subdivided our lead structures into four parts: the core (benzoxazole scaffold), carboxamide linker, and two variable groups R1 and R2 (Fig. 2A). We chose R1 and 2 for subsequent modifications to preserve the core nature of C40 and C127[87].

First, we substituted R1 with various moieties (Fig. 2B) and assessed all analogs for their *GRN* luciferase reporter activation efficacy from 10 nM to 30 μM over 48 h of treatment. To compare our analogs with each other, we chose the lowest concentration (3 μM) at which most analogs saturated the *GRN* luciferase reporter assay (Fig. 2C).

Through this SAR mapping, we identified analog A03 which achieved the highest *GRN* activation (indicated by our luciferase assay). A03 had the highest conformational freedom at R1 among all analogs, a key property at this position. Since A03 also presented reasonable solubility in vitro assays, we chose this R1-modification as our subsequent lead compound to continue our optimization process by substituting R2 (Fig. 2D). *GRN* activation was measured for all synthesized A03 analogs and compared after 24 h. (Fig. 2E). Most R2-substituted analogs (except A24, A28, A29) presented a similar or slightly higher activity compared to A03 at 3 μM (Fig. 2E).

During our optimization process, we evaluated the physicochemical properties of all analogs with a combination of experimental and computational approaches. These parameters describe the drug- and lead-likeness of a small molecule based on Lipinski's Rule of Five and predict important characteristics for efficacy, solubility, and permeability[88–90]. Typical parameters to address are logS (aqueous solubility) and logP (lipophilicity) values. Both parameters are critical for accessibility and distribution of the drug within the system[91–93]. Our effort to optimize our compounds in terms of activity led in most cases to molecules with a higher logP value while maintaining or increasing the logS value compared to C40 (Fig. 2F). However, high logP compounds can suffer from poor solubility and off-target effects. Identification of lower logP compounds that maintain high activity remains a desirable goal.

Along with logS and logP, we also examined topological polar surface area (TPSA), molecular weight (MW), fraction of sp3 hybridized carbon atoms, and number of rotatable bonds using SwissADME[94]. For a molecule to be able to cross the blood-brain barrier, lipophilicity, MW, flexibility and TPSA are crucial parameters[95,96]. We conducted principal component analysis (PCA) (Fig. 2G) and hierarchical clustering (Fig. 2H) with these predicted parameters (including predicted BBB penetrance ability, availability as a P-gp substrate (i.e., the likelihood of being pumped out of the cell by p-glycoprotein 1), and the measured luciferase assay activity) by using ClustVis[97] to identify physicochemical differences and similarities between our analogs. All analogs could be subdivided into two groups, either having high TPSA scores and logS values or high lipophilicity and flexibility. Since A03 presented increased solubility compared to C40 and C127, we decided to further assess analogs with similar characteristics.

## Compound analogs exhibit increased activity in vitro

To validate the results of the SAR-approach, twenty-six optimized analogs were evaluated in vitro over a dose range from 10 nM to 30 μM to establish luciferase activation (Supplementary Fig. 4a). Next, N2A cells were treated at four concentrations ranging from 1– 30 μM for induction of PGRN protein expression. Based on western blot data and in line with our SAR-approach results, the four compounds that demonstrated the most robust up-regulation of PGRN protein in N2A cells were A21, A39, A40, and A41 (Supplementary Fig. 4b, c). Encouragingly, all four compounds displayed a higher activity at identical doses when compared to the initial hits C40 and C127, with an average fold-induction of 2.5–3.3 x. We could not detect alterations in PGRN protein of intermediate lengths in our N2A treated cells (Supplementary Fig. 4D), although there is a possibility that our antibody was unable to bind to non-full length PGRN. Because PGRN is both a secreted and intracellular protein, we measured the relative levels of PGRN in N2A cell conditioned media and cell lysate after incubation with our compounds. We found that our compounds were able to increase PGRN in cell lysate but did not significantly increase extracellular PGRN in our in vitro model (Supplementary Fig. 5).

To determine if our four analogs were active in human cells, we treated U-373 MG Uppsala glioblastoma cells with 30 μM and 100 μM of each analog for 24 h (Fig. 3A, B) using 100 μM A03 as a positive control. We evaluated our analogs in U-373 cells at 30 and 100 μM to test whether our analogs were more active than our original compounds and if they had increased solubility in media which would allow substantial increased treatment concentrations. A21, A40, and A41 induced between 2 and 7-fold increases in PGRN protein between the two tested concentrations.

We next validated our analogs in human dermal fibroblast (HDF) cells from one *GRN* control (HDF X) and two patients with *GRN* haploinsufficiency (HDF Y and 2). HDF cells were treated for 24 h using A21 and A41 at two concentrations and then harvested for mRNA and protein analysis. We utilized an antibody that has been shown to react to full-length human PGRN, as well as granulin fragments 2 and 3 (Fig. 3C, D)[48]. Evaluation of treated cells revealed that A21 and A41 significantly increased *GRN* mRNA and full-length PGRN in both sets of *GRN* mutant cells and granulin fragments 2,3 in at least one set of *GRN* mutant cells (Fig. 3E). Further, A21 increased full length PGRN protein and *GRN* mRNA in a dose-dependent manner, while A41 reached its maximal effective concentration at 50 μM.

## Compound analogs are non-toxic, rescue Progranulin levels in vivo, and have slow decay in vitro

Following the selection of active analogs of interest and verification of enhanced activity in vitro, we evaluated selected compounds in vivo. Increased stability of our analogs in solution allowed us to inject A21, A39, A40, and A41 i.p. at 50 mg/kg into *Grn*[+/−] mice. Whole brain was harvested 24 h after injection and blotted for PGRN (Fig. 4A, B). Impressively, in Grn[+/−] mice A21 and A41 fully restored PGRN protein to *Grn*[+/+] levels which C40 had failed to do (Fig. 4B and 1F, G). Since we are unable to treat C40 at higher concentrations, we cannot verify if the analogs have an elevated activity level at higher concentrations compared to C40, but the ability of our analogs to fully restore PGRN levels could be due to increased activity, increased stability in solution and subsequent increased dosing, or both. Importantly, the small molecule SAHA which we had previously shown corrects PGRN levels in vitro had no effect on brain PGRN levels when injected IP. This demonstrates a major advantage of our compounds over the previously published small molecule SAHA and suggests that our compounds represent a more feasible chemical basis for *GRN*-FTD therapeutic development.

We next aimed to test the safety of prolonged exposure to our analogs. To accomplish this, we injected *Grn*[+/+] mice with 50 mg/kg A41 IP for 14 days. We measured the weights of A41 injected and vehicle injected mice daily and performed a complete blood count (CBC) analysis of all mice on day 14. We saw no noticeable changes in weight, grooming, energy, or overall external health after the 14-day treatment. In addition, CBC analysis did not detect any significant changes between A41 or vehicle treated mice. (Supplementary Fig. 5).

We further performed time course experiments (see Methods) to measure the persistence of the effect of our compounds

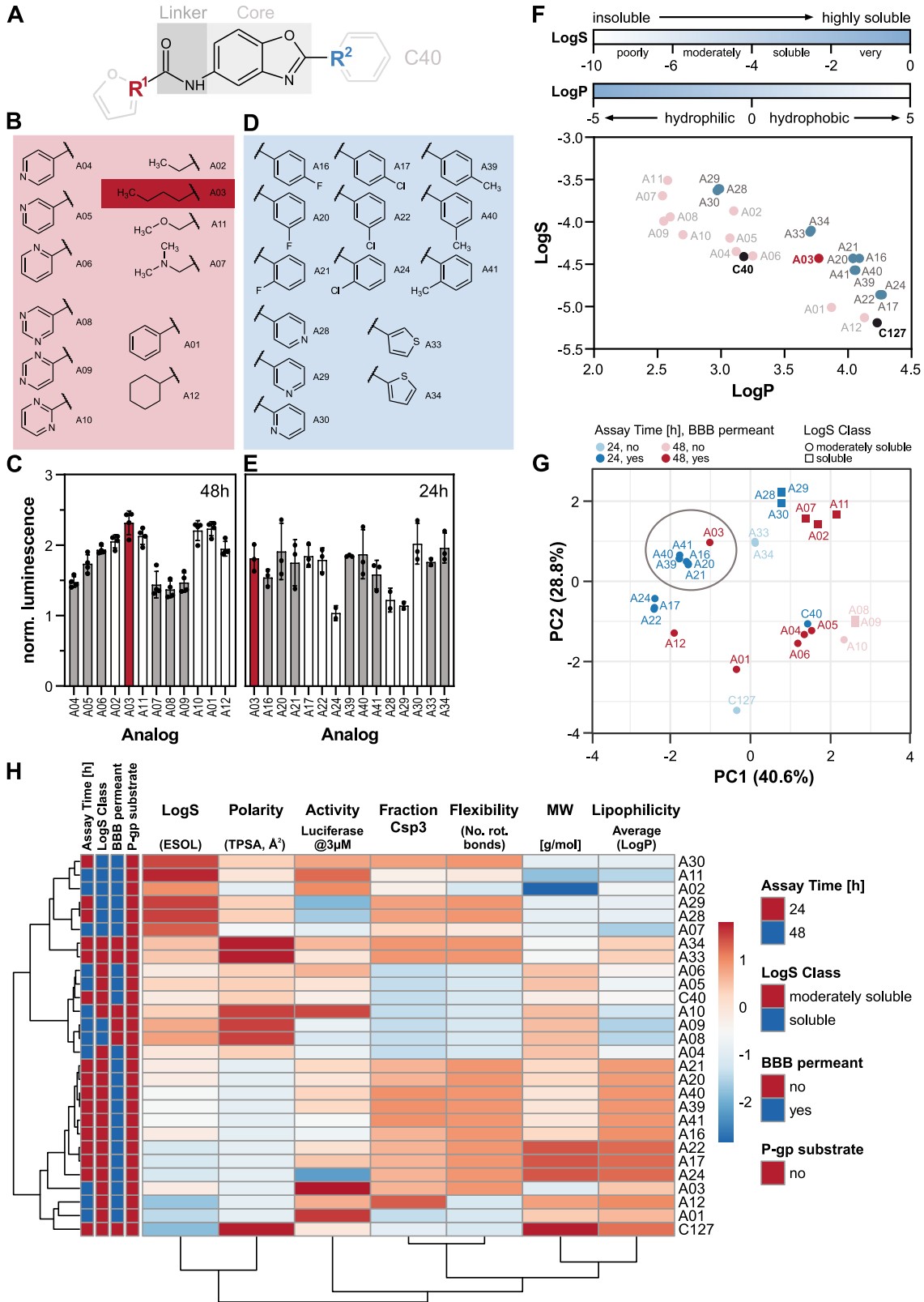

on PGRN levels and found *Grn* mRNA and PGRN protein remained elevated in vitro for 12 and 16 h, respectively, after drug removal (Fig. 4C, Supplementary Fig. 7a) and that PGRN protein level remained elevated in vivo up to 24 h after treatment (Fig. 4D, E, Supplementary Fig. 7b). Lastly, we performed PK analysis of A21 and A41 after i.p. injection with 25 mg/kg of either compound. Both analogs had decreased $C_{max}$ compared to C40, and

unlike C40, they did not preferentially partition into the brain; however, both A21 and A41 had much longer $t_{1/2}$ compared to C40 (119 and 98 min, respectively) (Supplementary Fig. 3d). Most importantly, in A21 and A41, we achieved our goals to successfully improve on the stability of C40 in solution and use these analogs to significantly increase PGRN levels in vivo when compared to C40.

**Fig. 2 | Structural optimization of the lead compound C40. A** Subdivision of lead compound C40 into core and linker region, R1 is the area altered in the first optimization round, R2 represents the area altered in the second optimization round after R1 was chosen. **B** Selected structures of functional groups used for R1 substitution. The dark red box indicates the moiety chosen in the first round of optimization based on the activity screen in. **C** Here, all analogs of the first optimization round of C40 were assessed for progranulin induction through the luciferase-based reporter assay after 48 h of treatment. Treatment concentration of 3 μM is shown normalized to the average value of DMSO controls ($n = 4$ biological replicates are summarized as mean ± SD). **D** In optimization round two, based on A03, R2 was substituted with shown functional groups. **E** Activity of new analogs were assessed via the luciferase-based reporter assay after 24 h of treatment and 3 μM treatment concentration normalized to the average value of DMSO controls is shown ($n = 3$ except for A24, A8, A29, A33, which are $n = 2$, biological replicates are summarized as mean ± SD). **F** Predicted LogS and LogP values of each analog of both optimization rounds as an indicator of solubility. Solubility scale of LogS and Lipophilicity scale of LogP are indicated, increasing blue color depicts increasing solubility. Predictions were made with SwissADME[94] and the consensus LogP was used. Initial compounds are shown in black, new lead analogs are shown in red, first round of molecules are colored in pink, second round of molecules are colored in blue. **G** Principal component analysis of predicted molecular features of all analogs including measured luciferase activity. Closest analogs to A03 are circled. PCA was conducted with ClustVis[97] and used parameters were predicted by SwissADME[94]. **H** Hierarchical clustering analysis of predicted and measured molecular features of all analogs conducted via ClustVis[97]. Rows are centered and variance scaling is applied. Correlation distance and average linkage is used for clustering both rows and columns.

## Progranulin-deficient cells demonstrate aberrant lysosomal proteomes

While our compounds demonstrated robust ability to enhance PGRN levels, it was imperative to test their ability to correct a disease-relevant cellular phenotype. However, this required assessment of the general effect of PGRN haploinsufficiency in our model systems. Most of the work performed on PGRN biology centers on PGRN homozygous models, and any heterozygous models (i.e. models that best genocopy actual FTD patients) have displayed modest, if any, pathological phenotypes[98,99]. However, previous transcriptomic and lipidomic studies of $GRN^{+/-}$ patient and mouse samples pointed towards altered lysosomes[23].

To exclusively investigate the lysosome of our model system, we performed a technique called LysoIP to specifically isolate lysosomes for downstream proteomic analyses[100–102]. Briefly, PGRN-deficient mouse embryonic fibroblasts (MEFs) or human-derived fibroblasts (HDFs) were engineered to express a lysosomal transmembrane protein, TMEM192, with a cytosolic 3xHA tag (Fig. 5A). These cells were then homogenized, intact lysosomes were isolated with anti-HA Dynabeads, and lysosomal proteins were extracted and subjected to tandem-mass-tag mass spectroscopy (TMT-MS) (Fig. 5B). LysoIP samples were checked for quality of extraction by western blot and *post hoc* Gene Ontology analyses of proteins obtained by TMT-MS (Fig. 5C, D, E)[103]. When compared to $Grn^{+/+}$ MEFs, $Grn^{+/-}$ and $Grn^{-/-}$ lysosomes demonstrated 40/765 (5.23%) and 42/94 (44.7%) differentially regulated proteins (Fig. 5F, Supplementary Fig. 8a, Supplementary Data 3, 4). Interestingly, when plotting the relative changes versus $Grn^{+/+}$ lysosomes for proteins detected in both $Grn^{+/-}$ and $Grn^{-/-}$ LysoIP analyses, a strong correlation ($p < 0.0001$) appears between the two genotypes with a slope of ~1.5, suggesting complete PGRN loss affects the same proteins to an ~1.5-times greater extent than the loss of one copy of PGRN (Supplementary Fig. 8b). This demonstrates a possible gene dosage dependent effect of PGRN on the lysosomal proteome.

When compared to $GRN^{+/+}$ HDFs, lysosomes from $GRN$ haploinsufficient patient-derived fibroblasts demonstrated 260/1218 (21.3%) differentially regulated proteins (Fig. 5g, Supplementary Data 5). The difference in number of proteins analyzed between LysoIP analyses is most likely due to a difference in total protein amounts used between TMT-MS runs. We further performed TMT-MS on HDF whole-cell lysates (Supplementary Data 6) and found a strong correlation in changes between $GRN^{+/+}$ and $GRN^{+/-}$ whole-cell lysate and LysoIP (Fig. 5H).

When comparing PGRN-haploinsufficient cell types, 16 proteins were significantly dysregulated (Fig. 5I). These proteins ensemble a broad functional spectrum, ranging from proteases (CTSA, CTSD, DPP7, and TPP1)[104–109], glycosidases (GAA, GLA, MAN2B1, MANBA)[110–112], and nucleases (DNASE2)[113] to regulators of lysosomal cholesterol/lipid levels (NPC1 and EPDR1)[114,115] and protein trafficking (IGF2R and SCAMP3)[116,117]. The wide range of lysosomal proteins altered in both

mouse and human PGRN-deficient cells points to the function of progranulin as a key regulator of lysosomal homeostasis across species.

## Progranulin-enhancing small molecules rescue the aberrant lysosomal proteome in FTD patient cells

Having discovered a robust lysosomal proteomic phenotype in PGRN haploinsufficient cells, we further tested the ability of one of our lead compounds, A41, to correct this phenotype.

First, we treated HDF cells from two separate $GRN$-FTD patients (i.e., PGRN haploinsufficient cells) with 50 μM A41 for 72 h and measured the levels of cleaved (i.e., lysosomal) TPP1, a protein significantly decreased in both LysoIP and whole-cell lysate TMT-MS datasets. Mutations in TPP1 are the cause of another type of neuronal ceroid lipofuscinosis, CLN2, underlining its importance for lysosomal homeostasis and function[105,107,118]. TPP1 was decreased by ~50% in untreated haploinsufficient HDFs compared to controls, and treatment with A41 led to a substantial increase in TPP1 protein levels (Fig. 6A, B). To determine how A41 was inducing increased lysosomal protein levels, we performed RT-qPCR to measure mRNA in untreated HDFs and HDFs treated for 24 or 72 h with 50 μM A41. We observed a time-dependent increase in $GRN$ and $TPP1$ transcript levels as well as in $DPP7$ (a second lysosomal protease decreased in PGRN haploinsufficient HDFs) (Fig. 6C). Previous work has demonstrated the lysosomal regulator transcription factor EB (TFEB) is a regulator of PGRN transcription[42]. Because A41 was affecting PGRN transcription as well as the transcription of other lysosomal proteins, we tested whether it was functioning by activating TFEB. To accomplish this, we measured the nuclear localization (i.e., activation) of TFEB-GFP in a stably expressing 293 T cell line upon addition of various concentrations of A41. Compared to DMSO control, A41 did not significantly increase the nuclear localization of TFEB until 100 μM, and this effect disappeared at 200 μM, suggesting there is no dose-dependence between A41 and TFEB activation (Fig. 6D, E). Furthermore, these results imply that A41 does not increase PGRN through TFEB since A41 is able to increase PGRN at levels lower than 100 μM.

To investigate the ability of A41 to correct lysosomal homeostasis, we treated $Grn^{+/-}$ MEF cells with 100 μM A41 for 24 h prior to LysoIP. This concentration is sufficient to increase PGRN back to $Grn^{+/+}$ levels in vitro. TMT-MS analysis of vehicle and A41 treated lysosomes demonstrated several key lysosomal proteins that are significantly altered in $Grn^{+/-}$ lysosomes are corrected upon treatment with A41 (Fig. 6F, J, Supplementary Data 7). Several of these proteins are key lysosomal catabolizing proteins that are mutated in various lysosomal storage disorders such as Tpp1, Manba, and Naga[107,119,120]. Interestingly, when comparing the relative changes versus $Grn^{+/+}$ for proteins detected in every MEF LysoIP-TMT-MS experiment ($Grn^{+/-}$ with DMSO and A41 and $Grn^{-/-}$) and performing principal component analysis (PCA), $Grn^{+/+}$, $Grn^{+/-}$, and $Grn^{-/-}$ all cluster separately, and A41 treated $Grn^{+/-}$ lysosomes cluster with $Grn^{+/+}$ lysosomes (Fig. 6G).

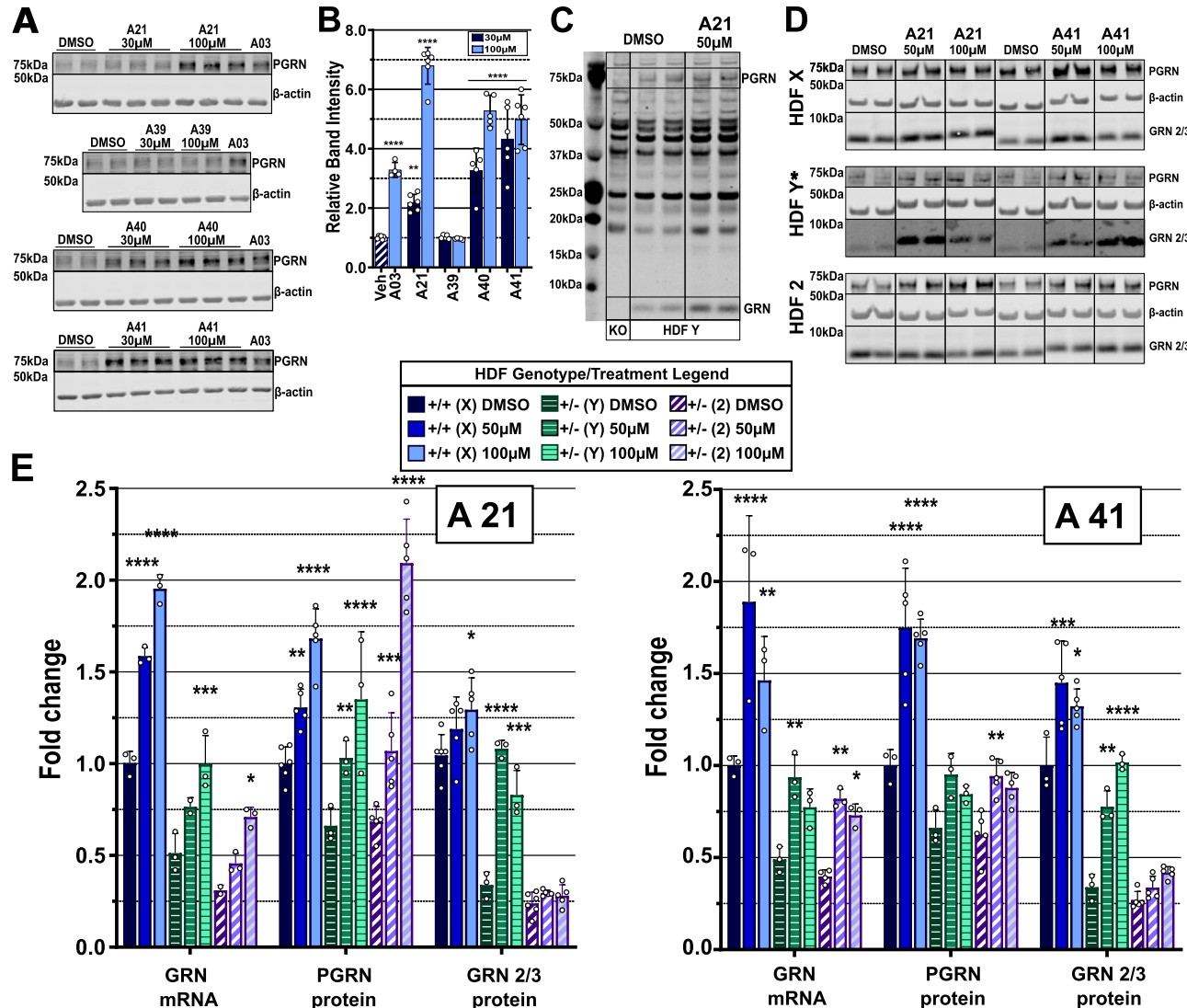

**Fig. 3 | Small molecules induce *GRN* mRNA and PGRN protein in human cells.**
**A** Western blots of U-373 cells treated with analogs and A03 ($n = 4$) positive control. **B** Quantification of western blots from section A. Both treated concentrations (30 μM and 100 μM) of A21 ($n = 6$), A40 ($n = 5$) and A41 ($n = 6$) increased PGRN by more than two-fold in U-373 cells. Cells showed no response to A39 ($n = 5$). Biological replicates are represented as mean values ± SD and an ordinary one-way ANOVA with Dunnett's multi comparisons test was used to calculate significance. * represents $p < 0.05$, **$p < 0.01$, ***$p < 0.001$, ****$p < 0.0001$ compared to DMSO treated control ($n = 7$). **C, D** Western blots of HDF cells generated from a WT control patient (HDF X) and 2 different *GRN* haploinsufficient patients (HDF Y and 2). HDF cells were treated with A21 and A41 and probed for full length PGRN and processed GRN fragments 2/3. (*HDF Y A21 and A41 were run on same gel so DMSO is same.

See source data for more information). **E** Quantification of HFD blots. Changes in *GRN* mRNA, PGRN protein and GRN fragments 2/3 protein quantified. A21 treatment (HDF X mRNA: all $n = 3$, PGRN and GRN2/3: DMSO $n = 6$, A21 treatment $n = 5$; HDF Y mRNA: DMSO $n = 3$, 50 μM $n = 2$, 100 μM $n = 3$, PGRN and GRN2/3: all $n = 3$; HDF (2) mRNA: DMSO $n = 2$, A21 treatment $n = 3$, PGRN and GRN2/3: all $n = 5$) and A41 treatment (HDF X mRNA: all $n = 3$, PGRN and GRN2/3: DMSO $n = 3$, A41 treatment $n = 5$; HDF Y all $n = 3$; HDF(2) mRNA: DMSO $n = 4$, A41 treatment $n = 3$, PGRN: DMSO and 100 μM $n = 5$, 50 μM $n = 6$, GRN2/3: all $n = 5$) are shown as mean values ± SD analyzed with two-way ANOVA with Dunnett's multiple comparisons test each patient cell set individually. * represents $p < 0.05$, **$p < 0.01$, ***$p < 0.001$, ****$p < 0.0001$ compared to DMSO treated control.

This demonstrates A41 is able to substantially correct the PGRN-deficient aberrant lysosomal proteome.

To further demonstrate the ability of A41 to correct the PGRN-deficient aberrant lysosomal proteome, we performed unbiased LysoIP TMT-MS analysis on *GRN*^+/− HDFs treated for 72 h with 50 μM A41 compared to *GRN*^+/+ HDFs. We then compared A41-treated *GRN*^+/− HDF quantitative spectral count changes (vs *GRN*^+/+ HDFs) to those in untreated *GRN*^+/− HDFs (vs *GRN*^+/+ HDFs) from our original LysoIP dataset and found significant differences in 287/547 (52.5%) proteins (Fig. 6H, K, Supplementary Data 8, 9). Of these, 147 proteins returned to *GRN*^+/+ levels, demonstrating substantial correction of the aberrant *GRN*^+/− lysosomal proteome with our PGRN-enhancing compound. PCA demonstrates the distinct clustering of *GRN*^+/+ and *GRN*^+/− proteomes as

well as the correction seen with A41 treatment (Fig. 6I), adding further evidence to the validity of our compounds as therapies for GRN-FTD.

## Discussion

A challenge of drug development is the discovery and optimization of compounds with the desirable pharmacological activity. In our effort to find potential PGRN-enhancing therapies for FTD, we aimed to identify agents that would be useful in developing a treatment for *GRN* deficiency-related FTD. One major goal was to obtain compounds that are BBB penetrant, since one of the greatest hurdles for the treatment of CNS disorders is the capacity of therapeutics to traverse the BBB. Fortunately, our initially identified, unaltered compounds, C40 and C127, were both BBB penetrant, albeit with divergent absorption

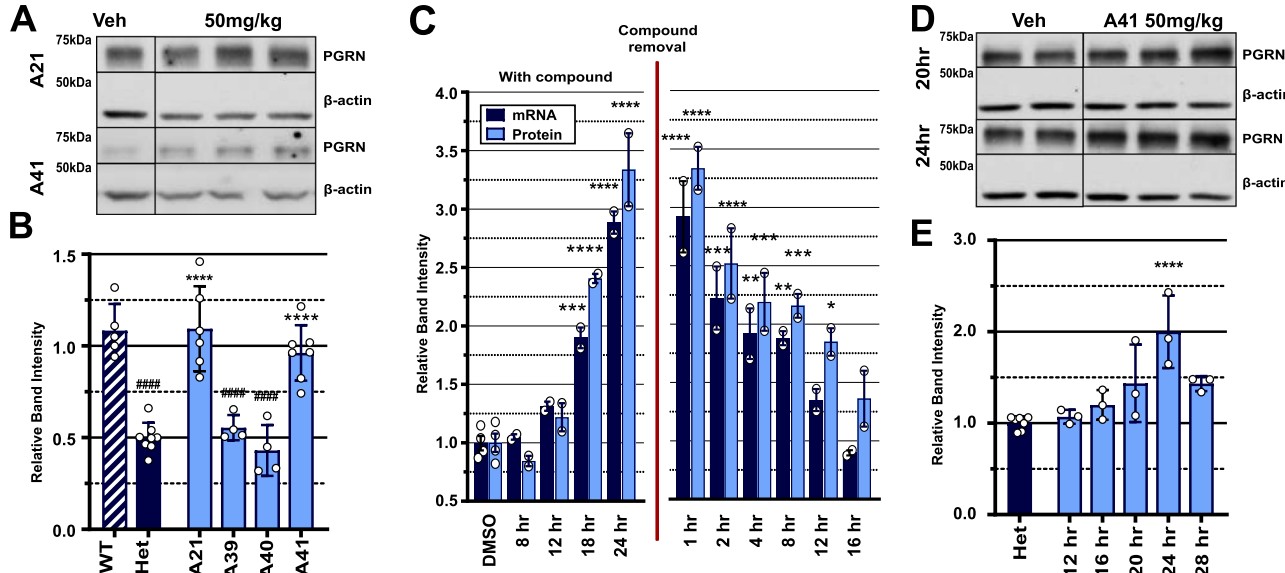

**Fig. 4 | Small molecule analogs lead to delayed induction of *Grn* mRNA and PGRN protein in vitro and in vivo. A, B** Western blot and quantification of *Grn*[+/−] mice treated i.p. with analog. A21 (*n* = 6) and A41 (*n* = 7) significantly increased PGRN protein in the brain compared to *Grn*[+/−] control. A21 and A41 also completely rescued *Grn*[+/−] PGRN protein back to *Grn*[+/+] levels while A39 (*n* = 4) and A40 (*n* = 4) had no effect. All biological replicates are summarized in mean values ± SD and * represents *p* < 0.05, **p* < 0.01, ***p* < 0.001, ****p* < 0.0001 significance with respect to *Grn*[+/−] control (*n* = 5); # represents *p* < 0.05, ##*p* < 0.01, ###*p* < 0.001, ####*p* < 0.0001 significance with respect to Grn[+/+] control (*n* = 5); ordinary one-way ANOVA with Dunnett's multiple comparisons test was used to determine significance. **C** N2A PRGN of A41 treatments at increasing time points (*n* = 2) with compound and after compound removal. Compound significantly increased PGRN

protein and *GRN* mRNA levels at 18 and 24 h. Statistically significant increased levels of PGRN protein persisted for 12 h after compound removal. Biological replicates are shown as mean values ± SD and ordinary one-way ANOVA with Dunnett's multiple comparisons was used to determine significance. * represents *p* < 0.05, **p* < 0.01, ***p* < 0.001, ****p* < 0.0001 in respect to DMSO treated control (*n* = 4). **D, E** Immunoblot and quantification of [*Grn*+/−] mice treated with 50 mg/kg A41 i.p. at increasing time points (*n* = 3 for each time point). A41 significantly upregulated PGRN protein at 20 and 24 h after treatment compared to controls (*n* = 6). (24 h data points included in quantification for Fig. 4B) Biological replicates are shown as mean values ± SD and ordinary one-way ANOVA with Dunnett's multiple comparisons was used to determine significance. * represents *p* < 0.05, **p* < 0.01, ***p* < 0.001, ****p* < 0.0001 in respect to DMSO treated control (*n* = 7).

properties. C40 was also able to significantly increase brain PGRN protein levels when given systemically.

To further optimize our compounds, we utilized SAR mapping to guide the synthesis of several analogs from the original hits. Of the resulting analogs generated, A21 and A41's high activity levels and improved solubility, secondary to introduction of a flexible alkyl moiety[92], enabled us to significantly increase PGRN levels in several human cell lines and in mouse brains.

When comparing the PK data of the analogs with C40, we found that all compounds were evenly distributed in the brain, but concentrations and persistence varied between the analogs and C40. Compared to C40, A41 resulted in lower overall concentrations but increased persistence in the brain. Flexibility, molecular weight, TPSA, and lipophilicity are critical parameters for BBB penetrance, and the literature indicates that the MW and number of rotatable bonds tend to be significantly lower for CNS active drugs compared to other drugs[121,122]. The TPSA value of A21 and A41 was 55 Å[2] and therefore lower than C40's TPSA value with 68 Å[2], supporting A21 and A41 BBB penetration. While our molecules also fit the MW criteria (400–600 Da), A03 and A17 - A41 contain 6 rotatable bonds, 1 greater than the upper limit of ideal[123–126]. Increasing flexibility can hinder membrane transit and might be the reason we observed less BBB penetrance for A21 and A41 compared to C40, even though the logP value of our analogs is higher compared to C40, which favors BBB crossing[127]. Regardless, both A21 and A41 increased PGRN protein levels to a significantly greater extent in vivo than did C40. A41 also did not alter mouse activity, did not induce a peripheral inflammatory cell response, nor did it decrease body weight after daily dosing for a continuous 14-day period.

Not only do our compounds increase PGRN levels, but they also correct the dysregulated lysosomal proteome in FTD patient cells. This

dysregulated proteome serves as a major phenotypic marker for FTD as several proteins associated with key lysosomal activities are decreased in *GRN*[+/−] HDFs. This included catalytic enzymes such as proteases, lipases, and nucleases as well as proteins that regulate cholesterol and lipid metabolism as well as lysosomal v-ATPase activity[128]. Treatment with our PGRN-enhancing molecules returned several of these proteins to *GRN*[+/+] levels, including GLB1, HEXA, and HEXB, key drivers of sphingolipid metabolism (Fig. 6E)[129,130]. Several species of gangliosides are increased in PGRN-deficient mouse and human samples, and loss of these enzymes may be, in part, responsible for this gangliosidosis[102,131].

*GRN*[+/−] lysosomes show ~30–50% reduction of several enzymes that, when deficient, cause lysosomal storage disorders[107,114,119,132–138]. While partial loss of any one of these enzymes is not sufficient to induce a lysosomal storage disorder, we hypothesize that the multitude of partial enzyme deficiencies (as seen in our PGRN deficient models) is sufficient to induce overt lysosomal dysfunction and eventually FTD. A41 was able to correct these lysosomal storage disease-related proteins back to *GRN*[+/+] levels. This promising result demonstrates the feasibility of our small molecule-based approach to not only increase PGRN levels in FTD patients, but to also correct the levels of several other disease-relevant lysosomal proteins.

In conclusion, we have identified four compounds, C40, C127, A21, and A41 which significantly upregulate PGRN on the transcriptional level in several human cell lines and correct PGRN protein levels in *Grn*[+/−] mouse brains. We have also identified an aberrant lysosomal proteome phenotype in PGRN-deficient models and have demonstrated the ability to correct this disease-relevant phenotype using our small molecules. Thus, these compounds are promising lead molecules for further drug development and improvement to gain a

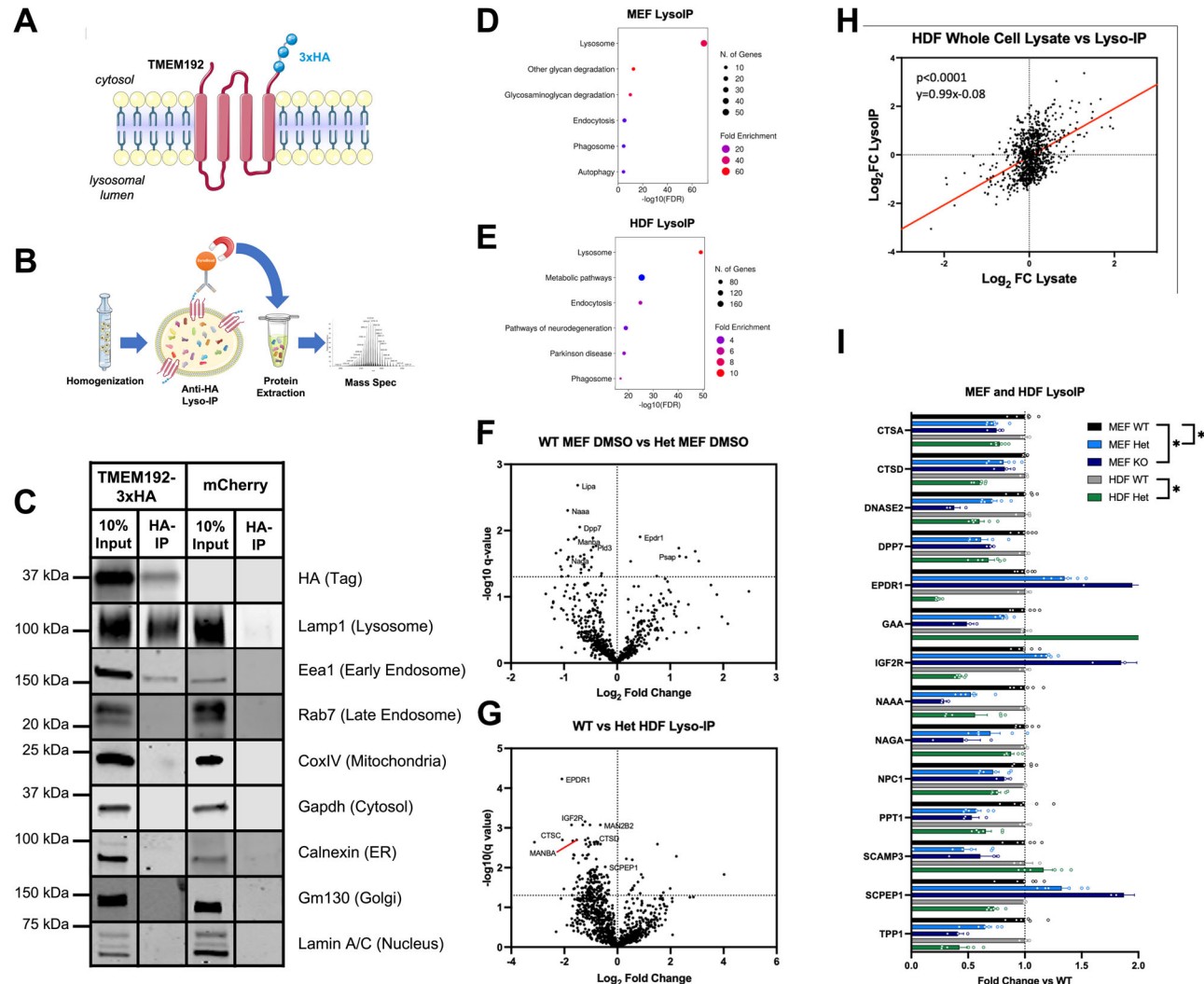

**Fig. 5 | LysoIP TMT-MS reveals loss of key lysosomal proteins in progranulin-deficient mouse and patient-derived fibroblasts. A** Diagram of 3xHA-tagged TMEM192 used for LysoIP. **B** Experimental design for LysoIP experiments. **C** Representative western blot of LysoIP experiment demonstrating specific isolation of lysosomes. Gene ontology of MEF (**D**) and HDF (**E**) LysoIP-derived proteins demonstrating enrichment of proteins under the KEGG term 'lysosome'. Volcano plots of LysoIP isolated lysosomal proteomes from $Grn^{+/-}$ MEFs ($n = 6$ $Grn^{+/+}$, $n = 6$ $Grn^{+/-}$) (**F**) and $GRN^{+/-}$ HDFs ($n = 3$ $GRN^{+/+}$, $n = 6$ $GRN^{+/-}$) (**G**). **H** $GRN^{+/-}$ vs $GRN^{+/+}$ fold changes of whole-cell lysate proteins correlate with fold changes in LysoIP-derived proteins ($p = 1.16 \times 10^{-50}$; simple linear regression). **I** LysoIP-derived proteins are significantly altered in mouse ($n = 6$ $Grn^{+/+}$, $n = 6$ $Grn^{+/-}$) and human ($n = 3$ $GRN^{+/+}$, $n = 6$ $GRN^{+/-}$) progranulin haploinsufficient cell types (*$q < 0.05$ Two-stage step-up Benjamini, Krieger, and Yekutieli Test). All error bars represent stardard error of the mean.

feasible and effective treatment of FTD caused by GRN haploinsufficiency in future.

## Methods

### Inclusion and ethics statement
All required permissions were obtained from institutional review boards.

### Experimental model and subject details
**Cell Lines**. All cell culture reagents and plates were purchased from Corning (Corning, NY, USA). Cells were maintained at 8.8% $CO_2$ at 37 °C. HEK 293 T cells were cultured in high glucose Dulbecco's Modified Eagle's Medium (DMEM) with 10% FCS and Pen/Strep and supplied from ATCC (CRL-3216). Neuro2A cells were cultured in high glucose DMEM with 10% FCS and Pen/Strep and supplied from ATCC (CCL-131). U-373 cells were grown in Minimum Essential Media with 10% FCS and Pen/Strep and supplied from ATCC (HTB-17). Non immortalized HDF cells were cultured in high glucose DMEM with 10% FCS plus supplemented L-glutamine and Pen/Strep and supplied from

the UCSF MAC. Mouse embryonic fibroblasts were generated as previously described[23] and were cultured in high glucose DMEM with 10% FCS plus supplemented L-glutamine and Pen/Strep.

### Generation of Grn mutant mice and animal husbandry
The *Grn* mutant mice were originally obtained from the Farese lab and were propagated for at least 50 generations in the Herz lab. Their generation is described in reference 40[40]. Animals were maintained on a mixed 129SvEv Bradley; C57BL/6 J background by heterozygous intercrossing. Wild type ($Grn^{+/+}$) control mice were obtained from the same crossings. Mice were genotyped by PCR using ear genomic DNA with primers (S1 5′-agtggggctggccacttct-3′, S2 5′-aagattcctcgctggga-catg-3′ and AS1 5′-gaatgctggtgtcagagggcc-3′)[40]. Animals were maintained on a 12 h light/12-h dark cycle and fed a standard rodent chow diet (Diet 7001; Harlan Teklad, Madison, WI) and water *ad libitum*. No sexual dimorphism of phenotype was observed. All procedures were performed in accordance with the protocols approved by the Institutional Animal Care and Use Committee of the University of Texas Southwestern Medical Center (IACUC APN 2015-101088-G).

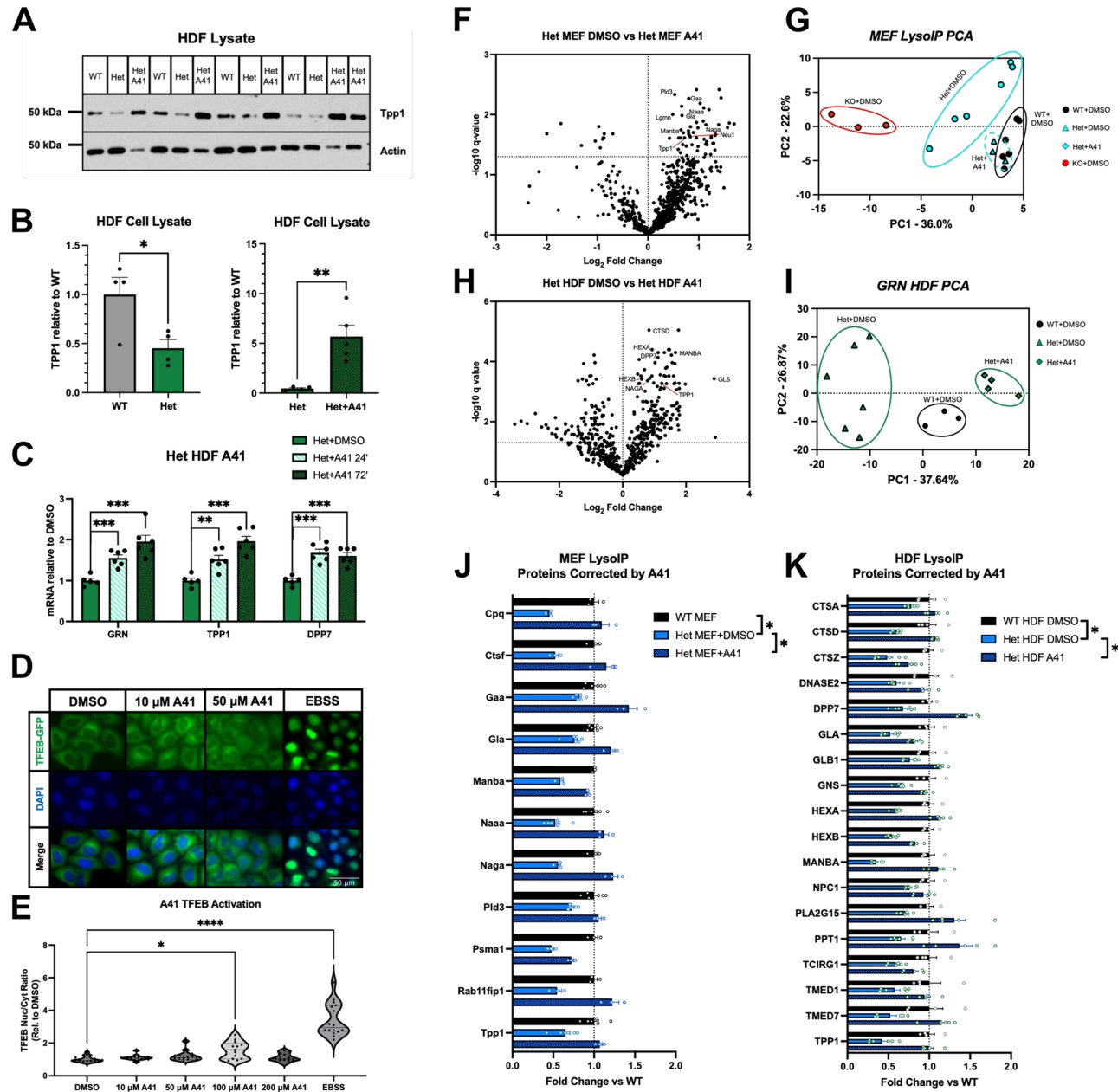

**Fig. 6 | Progranulin-boosting drugs correct aberrant lysosomal proteome.**
**A**, **B** Western blot validation demonstrates processed TPP1 levels in *GRN*+/− HDFs is reduced, and 24-h treatment with 50 μM A41 upregulates TPP1 levels ($n = 4$ *GRN*+/+, $n = 5$ *GRN*+/−, $n = 5$ *GRN*+/− + A41) (*$p = 0.03$; **$p = 0.005$ two-sided Student's *t*-Test). **C** RT-qPCR demonstrates upregulation of *GRN*, *TPP1*, and *DPP7* levels by 24- and 72 h treatment with 50 μM A41 ($n = 5$ *GRN*+/−, $n = 6$ *GRN*+/− + A41 24', $n = 5$ *GRN*+/− + A41 72') (**$p < 0.01$; ***$p < 0.001$; One-way ANOVA, Holm-Šídák correction). **D** Representative fluorescent microscopy images of TFEB-GFP (green) and DAPI (blue) in 293 T cells treated with various concentrations of A41 or EBSS (positive control). Micrographs represent 3 independent experiments. **E** Quantification of TFEB-GFP nuclear/cytoplasmic ratios (*$p = 0.014$; ****$p = 1.0 \times 10^{-15}$; One-way ANOVA, Holm-Šídák correction). **F** Volcano plot of lysosomal proteomes from *Grn*+/− MEFs treated for 24 h with 100 μM A41 compared to vehicle. **G** PCA depicting variations between DMSO treated *Grn*+/+, DMSO or A41 treated *Grn*+/−, and DMSO treated *Grn*−/− MEF lysosomal proteomes. **H** Volcano plot of lysosomal proteomes from *GRN*+/+ HDFs treated for 72 h with 50 μM A41 compared to vehicle. **I** PCA depicting variations between DMSO treated *GRN*+/+ and DMSO or A41 treated *GRN*+/− HDF lysosomal proteomes. **J** A41 treatment corrects the levels of several key lysosomal proteins in *Grn*+/− MEF lysosomes ($n = 6$ *Grn*+/−, $n = 6$ untreated *Grn*+/−, $n = 3$ A41 treated *Grn*+/−) (*$q < 0.05$; Two-stage step-up Benjamini, Krieger, and Yekutieli Test). **K** A41 treatment rescues the lysosomal expression of several proteins reduced in *GRN*+/− HDFs ($n = 5$ *GRN*+/−, $n = 6$ untreated *GRN*+/−, $n = 5$ A41 treated *GRN*+/−) (*$q < 0.05$; Two-stage step-up Benjamini, Krieger, and Yekutieli Test). All error bars represent standard error of the mean.

## Compounds

All resupplied from the 200,000 small molecule screen were purchased as powder from Chembridge (San Diego, California, USA) C15 (5185123); C40 (6328802); C41 (6161539); and Chemdiv (San Diego, California, USA) C19 (8009-7313); C66 (6143-0066); C105 (C276-0473); C107 (6131-0076); C127 (3864-1015). Analogs were synthesized in house.

## High throughput screening

The current UT Southwestern compound file is composed of 75,000 compounds purchased from ChemBridge Corporation, 100,000 compounds purchased from Chemical Diversity Labs and 22,000 compounds purchased from ComGenex, 1200 purchased from Tim-Tek, 1100 from Prestwick and the 450 compounds of the NIH Clinical Collection. This library also includes approximately 30,000 natural

products isolated from unique marine bacteria by Dr. John MacMillan (UC Santa Cruz). The TimTek compounds are "natural product-like" synthetic compounds and the Prestwick compounds are off-patent drugs.

In the screening assay, neuronally derived Neuro-2a (N2A) cells stably transfected with a luciferase reporter under the control of the GRN promoter were used to assess the effects of treatment on GRN activation (see Cenik et al. 2011[84] for methods and plasmid information). Cells were plated in 384 well plates using a BioTek MicroFlow™ (Agilent, Inc.). Cells were treated with the chemical library 4 h after plating and incubated for 24 h at 37 °C with 5% CO$_2$. Of importance, the compounds on each plate are plated to be in random order and not chemically similar. Compound concentration was set at 2.5 μM and final DMSO concentration in media was 1%. Column 1 contained the positive control sodium butyrate, and columns 2 and 23 contained the DMSO (internal) control. BrightGlo™ (Promega, Inc. E2620) was used to assay for luciferase levels. Proprietary pattern detection algorithms from Genedata (Screener™ v10 analysis suite) were used to perform quality control and normalize raw data for each plate. Raw data values for each plate were normalized using the equation:

$$Normalized\ values = 100*(Raw\ Values - Median\ of\ Test\ Population) \atop /Median\ of\ Test\ Population \tag{1}$$

Where the Test Population consists of library compounds in columns 3–22. The Robust Z-score for each compound was generated from the normalized, corrected data using the equation:

$$Robust\ Z\ Score = (Normalized\ Activity - Median\ of\ Neutral\ Controls) \atop /(Robust\ Stdev\ of\ Neutral\ Controls) \tag{2}$$

Robust Z Score was used to select primary hits. Any compound with a score of 3 or higher was considered a primary hit. To choose suitable compounds to pursue from our primary hits, a two-point activity (normalized activity to assay controls) was used with endogenous levels (DMSO treated) set at 0 and the positive control set at 100. Sodium butyrate was used for the positive control at a concentration of 9 μM and induced a two-fold increase in luciferase activation compared to DMSO treated cells. Compound toxicity was assessed against their activity in 12 NSCLC cell lines. CellTiterGlo™ (Promega, Inc., G7572) was used to identify changes in ATP levels compared to DMSO control due to accelerated cell growth or death.

### In vitro toxicity assay
All cells were grown in 96 well plates with 4 replicates for each treatment. CellTiterGlo™ (Promega, Inc., G7572), which uses ATP as an indicator of metabolically active cells and estimates cell viability based on ATP levels in cells, was utilized to assess toxicity. The assay consumes ATP to produce a luciferase signal which is proportional to the amount of ATP in the cultured cells.

**In vitro compound treatment.** Neuro2A, and U-373 cells were grown in 6 well plates for harvesting. HDF cells were grown in 100 mm plates for harvesting. All cells were treated at a final concentration of 1% DMSO with or without compound over a specific time period before harvest as noted in text.

**Secretion Assay.** N2A cells were plated, and media was replaced with serum free media when cells were ˜ 90% confluent. Cells were then treated with 0 μM, 10 μM, or 30 μM of either A21 or A41 for 24 h before conditioned media was saved for analysis and cells were lysed with 150 μL of lysis buffer. 500 μL of media was concentrated to 100 μl (i.e., 5x concentration) using Amicon ultracentrifugation concentration tubes with a 3 kDa cutoff (EMD Millipore, UFC500308). Lysate and concentrated conditioned media were then analyzed by western blot.

### ICV cannulation
All compounds were dissolved in DMSO at a concentration of 100 μM and this solution was further diluted in artificial cerebrospinal fluid (ACSF) (128 mM NaCl, 2.5 mM KCl, 0.95 mM CaCl$_2$, 1.99 mM MgCl$_2$). The final concentration of all compounds, when added to the pump, but pre-infusion, was 10 μM in 10% DMSO and 90% ACSF. The day before surgery, 1007D osmotic pumps (Alzet, Cupertino, CA, USA, 0000290) (flow rate: 0.5 μL/h) were filled with the compound solution, connected via tubing to a cannula [tubing and cannula provided from brain infusion kit 3 (Alzet, Cupertino, CA, USA, 0008851)], placed in 0.9% sodium chloride irrigation, and primed overnight in a water bath at 37 °C.

Animals were anesthetized and placed in a stereotaxic apparatus. Cannula and pump were implanted as described for ICV implantation in the manufacturer's surgical procedures (Alzet) at the coordinates 0.3 mm posterior and 1.25 mm lateral of bregma at a depth of 3.5 mm. 4/0 blue monofilament nylon non-absorbable sutures were used for wound closure. Carprofen was given as needed for pain management during and after cannulation. Implantation was conducted on 72 three-month-old (± 5 days), male *Grn* wild-type or mutant mice on a mixed 129SvEv Bradley; C57BL/6 J background.

Animals were sacrificed and harvested 7 days after implantation. Mice were deeply anesthetized with isoflurane and transcardially perfused with phosphate buffered saline (PBS). The whole brain was removed from the skull, hemispheres were separated and both hemispheres were snap frozen in liquid nitrogen and stored at −80 °C.

### I.P. injections
66 three-month-old (± 5 days), male *Grn* wild-type or mutant mice were injected with the following compounds in their respective vehicles. C40 was delivered at 10 mg/kg, formulated as 10% DMSO, 5% Tween 80, 85% of a 30% solution of HPβCD. A21 was delivered at 50 mg/kg formulated in straight DMSO. A39, A40, A41 and SAHA were delivered at 50 mg/kg in 5% DMSO, 45% PEG, 50% HPβCD (600 mg/ml). C40 and A41 treatments were delivered at 0.2 mL per 25 grams of mouse. A21 was delivered at 0.04 mL per injection. Mice were sacrificed and tissue harvested as described in ICV methods. Time points for harvest as noted in data sets.

### Immunoblotting
All cells were rinsed with PBS then harvested in RIPA buffer (10 mM Tris-HCL pH 8.0, 140 mM NaCl, 0.1% sodium deoxycholate, 0.1% SDS, 1% triton X-100, 1 mM EDTA, 0.5 mM EGTA). The lysate was then cleared by centrifugation at 21,000 × *g* for 15 min at 4 °C. Brain hemispheres were lysed in RIPA buffer (50 mM Tris-HCL pH 7.4, 150 mM NaCl, 0.25% sodium deoxycholate, 1% NP-40, 1 mM EDTA) and homogenized for 20 s in a 1 mL glass homogenizer. The lysate was cleared by centrifugation at 21,000 × *g* for 20 min at 4 °C. Protein concentrations were conducted on all sample sets using DC protein assay kit (Bio-Rad, Hercules, CA, USA). Cells were denatured in a sample buffer of 50 mM TCEP, 0.4% Orange G, 40% glycerol, 8% LDS, 125 mM Tris-HCl (pH 6.8) while tissue used Tris-glycine SDS 2x sample buffer (Life Technologies, Carlsbad, CA, USA, LC2676) plus 5% β-mercaptoethanol. All samples were loaded at 25 μg per well on a 10% Tris-glycine gel and separated by SDS-PAGE. Cell samples were transferred to nitrocellulose membranes using Trans-Blot Turbo Transfer System (Bio-Rad) while tissue samples were transferred to polyvinylidene difluoride membranes (PVDF). Both were blocked for 1 h in 5% BSA in TBS with 0.1% Tween 20 for 1 h. Membranes were incubated with proper primary antibody solutions on a rocking shaker overnight at 4 °C. Nitrocellulose membranes were incubated in fluorescent secondary for 1 h and visualized using the Odyssey Imaging System (Li-cor, Lincoln, NE, USA). PVDF membranes

were incubated for 2 h on a rocking shaker with ECL-conjugated secondary antibodies. The antibodies were visualized by chemiluminescence and the membranes were exposed to blue x-ray film for 15–180 s at room temperature. All band signals were quantified and normalized to loading control. Primary antibodies: Anti-GRN 1:1,500 dilution, (sigma HPA008763, Lot 000043984); Anti-granulin 1:3,000 (Abcam ab252834, Lot GR334867-1); Anti-beta actin 1:5,000–1:20,000 dilution, (Abcam ab8227, GR342990-1); Anti-HA 1:1000 dilution (Cell Signaling 3724 S, Lot 10); Anti-mouse CD107a 1:1000 dilution (BD Biosciences 553792, Lot 0000055197); Anti-EEA1 1:1000 dilution (Cell Signaling 3288 S, Lot 8); Anti-Rab7 1:1000 dilution (Cell Signaling 9367 S, Lot 3); Anti-GM130 1:1000 dilution (Thermo Fisher MA5-35107, Lot XD3555932); Anti-COX IV 1:1000 dilution (Cell Signaling 4850 S, Lot 10); Anti-Lamin A + C 1:1000 dilution (Abcam ab133256, Lot 7); Anti-Calnexin 1:12,000 dilution (Cell Signaling 2679 S, Lot 03011836); Anti-TPP1 1:100 dilution (Santa Cruz sc-393961, Lot L1621); Secondary antibodies: HRP Anti-rabbit IgG 1:2000-1:20,000 dilution (Cytivia NA9341ML, Lot 18111932); HRP anti-rat IgG 1:5,000 dilution (Biolegend 405405, Lot B364032); IRDye 800CW anti-rabbit IgG 1:20,000 dilution (Li-cor 926-32213, Lot D30627-15); IRDye 800CW donkey anti-mouse IgG 1:20,000 dilution (Li-Cor 926-32212, Lot C61116-02); IRDye 680RD donkey anti-rabbit IgG 1:20,000 dilution (Li-Cor 926-68073, Lot D30627-15); IRDye 680RD goat anti-rat IgG 1:20,000 dilution (Li-Cor 926-68076, Lot AB_10956590).

## Murine microsome S9 fractions

Compound was incubated with Murine S9 (Lot VLH) fraction and Phase I (NADPH Regenerating System) cofactors for 0–240 min. Reactions were quenched with 0.5 mL (1:1) of methanol containing 0.2% formic acid and 50 ng/ml IS (IS final conc. = 25 ng/ml). Samples were vortexed for 15 s, incubated at RT for 10 min and spun for 5 min at $500 \times g$. Supernatant (1 mL) was then transferred to an Eppendorf tube and spun in a tabletop, chilled centrifuge for 5 min at $16,000 \times g$. Supernatant (800 μL) was transferred to an HPLC vial (w/out insert). Analyzed by Qtrap 4000 mass spectrometer.

*MS parameters (4000 Qtrap:)*

Method: compound pos + NB IS 110916

Ion Source/Gas Parameters: CUR = 45 CAD = med, IS = 5000, TEM = 700, GS1 = 70, GS2 = 70.

Buffer A: Water + 0.1% formic acid; Buffer B: Acetonitrile + 0.1% formic acid; flow rate 1.5 ml/min; column Agilent C18 XDB column, 5 micron packing 50 × 4.6 mm size; 0–1.0 min 3% B, 1.0–1.5 min gradient to 100% B, 1.5–3.0 min 100% B, 3.0–3.1 min gradient to 3% B, 3.1–4.0 3% B; IS: N-benzyl-benzamide (Sigma-Aldrich, transition 269.1 to 169.9); Compound transition 335.137/251.0.

**Plasma stability.** Compound (2 μM final concentration) was incubated with murine plasma and saline for 0-1440 min. Reactions were quenched with 200 μl (1:1) of methanol containing 0.2% formic acid and 50 ng/ml IS. Samples were vortexed for 15 s, incubated at RT for 10 min and spun for 5 min at $16,000 \times g$. Supernatant was then transferred to an Eppendorf tube and spun in a tabletop, chilled centrifuge for 5 min at $16,000 \times g$. Supernatant was transferred to an HPLC vial (w/ insert). Analyzed by Qtrap 4000 mass spectrometer.

*MS parameters (4000 Qtrap:)*

Method: C40 pos + NB IS 110916

Ion Source/Gas Parameters: CUR = 45 CAD = med, IS = 5000, TEM = 700, GS1 = 70, GS2 = 70.

Buffer A: Water + 0.1% formic acid; Buffer B: Acetonitrile + 0.1% formic acid; flow rate 1.5 ml/min; column Agilent C18 XDB column, 5 micron packing 50 × 4.6 mm size; 0–1.0 min 3% B, 1.0–1.5 min gradient to 100% B, 1.5–3.0 min 100% B, 3.0–3.1 min gradient to 3% B, 3.1–4.0 3% B; IS: N-benzyl-benzamide (sigma -aldrich, transition 269.1 to 169.9); Compound transition 305.159/237.1.

## Caco2 assay

Caco2-cell monolayer was cultured for 26 days. Monolayer was rinsed with HBSS 0.5% FBS 3 times before 10 μM compound in HBSS 0.5% FBS solution (0.5 mL to apical side for A to B transport and 1.5 ml to basolateral side for B to A transport) was added. 1.5 ml of plain HBSS 0.5%FBS was added to basolateral side for A to B transport and 0.5 ml of plain HBSS0.5% FBS was added to apical side for B to A transport.

At 30, 60, 90, 120 min post compound addition, 1000 μl was removed from basolateral side of A to B sample well; 300 μl was removed from apical side of B to A analysis well. Removed media was replaced with blank HBSS 0.5% FBS to ensure sink conditions.

200 μl MeOH with 15 ng/ml benzylbenzamide as an internal standard (IS) and 2 mM Ammonium acetate, 0.15% formic acid was added to 100 μl sample from each time point. Sample was vortexed for 15 s, incubated 10 min RT, spun $16,000 \times g$ for 5 min, and supernatant was analyzed by LC-MS/MS using Shimadzu Prominence LC and AB Sciex 4000Trap.

Caco2 Controls Prop Nal Quin Cim 031916.dam for control compound

*MS parameters (4000 Qtrap:)*

Control Compounds: Ion Source/Gas Parameters: CUR = 45, CAD = Medium, IS = 5500, TEM = 400, GS1 = 50, GS2 = 50.

Buffer A: $dH_2O$ + 2 mM ammonium acetate + 0.1% formic acid; Buffer B: MeOH + 2 mM ammonium acetate + 0.1% formic acid; flow rate 1.5 ml/min; column Agilent C18 XDB column, 5 micron packing 50 ×4.6 mm size; 0–1.5 min 97% A, 1.5–2.5 min gradient to 100% B, 2.5–3.5 min 100% B, 3.5–3.6 min gradient to 97% A, 3.6–4.5 min 97% A; IS: n-benzylbenzamide (sigma -aldrich, lot #02914LH, transition 212.1 to 91.1). Compound transitions - see information above.

**In vivo I.P.** Twenty-one six week old female CD-1 mice were dosed i.p. with compound. C40 was delivered at 10 mg/kg, 0.2 ml/mouse formulated as 10% DMSO/5% Tween 80/85% of a 30% solution of HPβCD. C127 was delivered at 10 mg/kg C127, 0.2 ml/mouse formulated as 10% DMSO/10% Cremophor EL/80% D5W (5% dextrose in dH20, pH 7.4). A21 was delivered at 30 mg/kg, 0.04 lmL/mouse formulated in straight DMSO. A41 was delivered at 25 mg/kg, 0.2 ml/mouse formulated as 5% DMSO, 45% PEG, 50% HPβCD (600 mg/ml). Whole blood and brains were harvested. Plasma was processed from whole blood by centrifugation of the ACD treated blood for 10′ at $10,000 \times g$ in a standard centrifuge. Brains were weighed and snap frozen in liquid nitrogen.

MS parameters (3200 Qtrap:)

Method: Compound + pos IS 113016Ion Source/Gas Parameters: CUR = 45 CAD = med, IS = 5000, TEM = 700, GS1 = 70, GS2 = 70.Buffer A: Water + 0.1% formic acid; Buffer B: Acetonitrile + 0.1% formic acid; flow rate 1.5 ml/min; column Agilent C18 XDB column, 5 micron packing 50 × 4.6 mm size; 0–1.0 min 3% B, 1.0–1.5 min gradient to 100% B, 1.5–3.0 min 100% B, 3.0–3.1 min gradient to 3% B, 3.1–4.0 3% B; IS: N-benzyl-benzamide (sigma -aldrich, transition 269.1 to 169.9); Compound transition 305.159/237.1.

**In vivo P.O..** Twenty-one six week old female CD-1 mice were dosed p.o. with 10 mg/kg C40, 0.2 ml/mouse formulated as 10% DMSO/5% Tween 80/85% of a 30% solution of HPbCD. Whole blood and brains were harvested. Plasma was processed from whole blood by centrifugation of the ACD-treated blood for 10′ at $10,000 \times g$ in a standard centrifuge. Brains were weighed and snap frozen in liquid nitrogen.

MS parameters (3200 Qtrap:)

Method: C40 +pos IS 113016

Ion Source/Gas Parameters: CUR = 45 CAD = med, IS = 5000, TEM = 700, GS1 = 70, GS2 = 70.

Buffer A: Water + 0.1% formic acid; Buffer B: Acetonitrile + 0.1% formic acid; flow rate 1.5 ml/min; column Agilent C18 XDB column, 5 micron packing 50 × 4.6 mm size; 0–1.0 min 3% B, 1.0–1.5 min

gradient to 100% B, 1.5–3.0 min 100% B, 3.0–3.1 min gradient to 3% B, 3.1–4.0 3% B; IS: N-benzyl-benzamide (Sigma-Aldrich, transition 269.1 to 169.9); Compound transition 305.159/237.1.

## Mouse complete blood count

Whole mouse blood was collected in K3-citrate tubes and inverted several times. Samples were kept at room temperature and measured on IDEXX Procyte Analyzer with tube adapter.

## Quantitative real-time PCR

Total cellular RNA was isolated from cells using Trizol (Invitrogen, Waltham, MA, USA). and diluted to 100 ng/10 μl then reverse-transcribed into cDNA using a High-Capacity cDNA Reverse Transcription Kit. Real-Time PCR was performed using PowerUp SYBR Green Master Mix, and analysis was conducted on a ViiA 7 (Thermo-Fisher, Waltham, MA, USA). Cyclophilin and h36B4 were used for normalization.

## TFEB activation assay

293 T cells stably expressing TFEB-GFP were plated on coverslips and treated for 24 h with 0, 10, 50, 100, or 200 μM A41 in 1% (v/v) DMSO. For the positive control, cells were treated with DMSO only for 24 h and media was replaced by Earle's balanced salt solution for 4 h. Cells were fixed with 4% paraformaldehyde for 10 min room temperature, washed three times for 5 min with PBS (second wash included 1:10,000 DAPI), and mounted on microscope slides with vectamount. Images were taken on an Axioplan 2 microscope, and images were analyzed by ImageJ.

## Lentivirus production

For lentivirus production, 70% confluent 15 cm plates of HEK 293 T cells were co-transfected with 6.75 μg psPAX2, 2.25 μg pMD2.g, and 9 μg the individual construct encoding plasmids. Plasmids were added in a 1:3 ratio of Fugene 6 in 900 μl OptiMEM media. The media was replaced after 12–16 h. Viral particle-containing media was collected, and cell debris spun down. The viral particles were frozen at -80°C. Cells were infected with conditioned media, and media was replaced after 24 h.

## LysoIP

MEFs and HDFs were grown to ~80% confluency in 15 cm plates and infected with lentivirus expressing either TMEM192-3xHA-IRES-mCherry or IRES-mCherry alone. Media was replaced after 24 h, and 48 h post-infection, cells underwent LysoIP. Briefly, cells were washed with cold PBS and scraped into 15 ml conical Falcon tubes in 10 mL KPBS (136 mM KCl and 10 mM $KH_2PO_4$, pH 7.2). HDFs were pooled 3-4 plates/sample. Cells were spun down at 200 × g 4 °C for 5 min. Liquid was decanted, and cells were resuspended in 1 ml KPBS supplemented with protease and phosphatase inhibitors (Roche). All further steps were performed in a cold room. Cells were "vacuum homogenized" by pulling them into a 5 ml syringe, expelling all air, covering the syringe tip with parafilm, pulling the plunger to create a vacuum inside the syringe, and releasing the plunger to strike the cells against a hard surface. This was repeated 6-8 times/sample. Samples were then spun at 2000 × g 4 °C for 2 min, and the supernatant was added to 50 μl anti-HA Dynabeads (ThermoFisher, Waltham, MA, USA, 88837) pre-washed in KPBS. Samples were incubated with the beads for 10 min on a rotator at 4 °C. Beads were washed 3x with KPBS+protease and phosphatase inhibitors, being moved to fresh tube on third wash. Proteins were eluted from beads by incubation with 100 μl triton-X lysis buffer (50 mM Tris-HCl, pH 7.4, 150 mM NaCl, 1 mM EDTA, 1% (vol/vol) Triton X-100, protease and phosphatase inhibitors) on ice for 10 min. Beads were removed from solution, and debris was spun down at 17,000 × g 4 °C for 10 min. All steps were performed with low retention tubes (ThermoFisher, Waltham, MA, USA, 3451).

## Tandem mass tag mass spectrometry

Samples were dried for 30 min in a SpeedVac, after which 40 μl of 5% SDS was added to each. Samples were then reduced with TCEP and alkylated with iodoacetamide in the dark. Each sample was loaded onto an S-Trap Micro (Protifi), following which 2 μg of trypsin (Pierce) was added and allowed to digest overnight at 37 °C. Peptides were eluted and dried in a SpeedVac, then were reconstituted in 21 μl of 50 mM TEAB buffer. Samples were then each labeled with 4 μl of TMTpro 16plex reagent (ThermoFisher, Waltham, MA, USA, A44520) and quenched with 2 μl of 5% hydroxylamine and combined in equal peptide amount based on NanoDrop A205 reading. These mixtures were dried in a SpeedVac and reconstituted in 2% acetonitrile, 0.1% TFA buffer.

The TMTpro 16plex sample was injected into an Orbitrap Fusion Lumos mass spectrometer coupled to an Ultimate 3000 RSLC-Nano liquid chromatography system. The sample was injected onto a 75 μm i.d., 75 cm long EasySpray column (ThermoFisher, Waltham, MA, USA, ES900) and eluted with a gradient from 0–28% buffer B over 180 min. Buffer A contained 2% (v/v) ACN and 0.1% formic acid in water, and buffer B contained 80% (v/v) ACN, 10% (v/v) trifluoroethanol, and 0.1% formic acid in water. The mass spectrometer operated in positive ion mode with a source voltage of 1.5 kV and an ion transfer tube temperature of 275 °C. MS scans were acquired at 120,000 resolution in the Orbitrap and top speed mode was used for SPS-MS3 analysis with a cycle time of 3 s. MS2 was performed with CID with a collision energy of 35%. The top 10 fragments were selected for MS3 fragmentation using HCD, with a collision energy of 55%. Dynamic exclusion was set for 25 s after an ion was selected for fragmentation.

Raw MS data files were analyzed using Proteome Discoverer v2.4 (ThermoFisher, Waltham, MA, USA), with peptide identification performed using Sequest HT searching against the mouse protein database from UniProt. Fragment and precursor tolerances of 10 ppm and 0.6 Da were specified, and three missed cleavages were allowed. Carbamidomethylation of Cys and TMTpro labelling of N-termini and Lys sidechains were set as a fixed modification, with oxidation of Met set as a variable modification. The false-discovery rate (FDR) cutoff was 1% for all peptides.

## Analysis of TMT-MS results

For each TMT-MS run, at least one LysoIP sample from cells expressing IRES-mCherry only was included as a negative control. Only proteins whose raw abundance in every sample was >1.5x the abundance of mCherry were considered for further analysis. Raw abundance of mCherry control was subtracted from the raw abundance of each sample, and each sample was normalized to the raw abundance of Lamp1. Progranulin-deficient cells have an increased number of lysosomes[23], thus normalizing to a non-lysosomal standard would artificially increase the level of lysosomal proteins in progranulin-deficient samples. Gene Ontology analysis was performed using ShinyGO 0.77 (http://bioinformatics.sdstate.edu/go/).

## Quantification and statistical analysis

Data analysis was conducted using ImageJ (NIH) and Odyssey Imaging Systems (Li-Cor). GraphPad Prism was used for all statistical analysis except PK analysis, which used Microsoft Excel. Data are displayed as means ± standard deviation (SD). Ordinary one-way ANOVA was used to compare multiple groups with common control(s). Repeated measured two-way ANOVA was used for weight analysis. For TMT-MS analysis, fold changes were compared by multiple t-tests followed by two-stage step-up Benjamini, Krieger, and Yekutieli correction. To show statistical significance, * or # represents a $p < 0.05$. Replicate number shown in figure graphs.

| Primer Sequences | |
|---|---|
| Mouse *Grn* Forward | 5′-GTCCTGGGAGCCAGTTTGAA-3′ |
| Mouse *Grn* Reverse | 5′-CATCCCCACGAACCATCAAC-3′ |
| Mouse Cyclophilin Forward | 5′-TGGAGAGCACCAAGACAGACA-3′ |
| Mouse Cyclophilin Reverse | 5′-TGCCGGAGTCGACAATGAT-3′ |
| Human *GRN* Forward | 5′-GAGATGTCCCCTGTGA-TAATGTCA-3′ |
| Human *GRN* Reverse | 5′-CCACTCCCCAGACGTGAGTT-3′ |
| Human *TPP1* Forward | 5′-GGTGGCTTCAGCAATGTGTTCC-3′ |
| Human *TPP1* Reverse | 5′-GAAGTAACTGGATGGTGGCAGG-3′ |
| Human *DPP7* Forward | 5′-CACCATCCAGTTACTTCAATGC-3′ |
| Human *DPP7* Reverse | 5′-CTGACCCTCCACTTCTTCATTC-3′ |
| Human *h36B4* Forward | 5′-TGCATCAGTACCCCATTCTATCA-3′ |

## Compound synthesis

HRMS data for A21, A39, A40, A41 found in supplemental materials

Synthesis of 2-substituted 1,3-benzoxazol-5-amine derivatives

**2,4-diaminophenol** M = 124.14 g/mol + **thiophenecarboxylic/ benzoic acid derivate** → (PPA, 180°C, 3-5 h) → **2-phenylbenzo[*d*]oxazol-5-amine derivate**

This synthesis was partly performed as described in Chancellor et al., 2011[139].

4.3 g polyphosphoric acid (PPA) was heated in a 5 ml glass vial to 110 °C under stirring. 0.3 g of respective thiophenecarboxylic/ benzoic acid derivative and 1 mol equiv. of 2,4-diaminophenol were added simultaneously to the heated PPA. While stirring, the mixture was heated to 180 °C until no starting material was left. LC-MS was used for reaction control.

After complete conversion, the mixture was cooled to room temperature and to obtain the final product, water was added into the reaction vial to precipitate the crude solid. In some cases, the solid was recrystallized with hot ethanol for further purification.

Synthesis of *N*-(2-substituted 1,3-benzoxazol-5-yl)-amide derivatives

**2-substituted benzo[*d*]oxazol-5-amine derivate** + **carboxylic acid** → (HATU, DIPEA, r.t., DMF) → ***N*-(2-substituted benzo[*d*]oxazol-5-yl)-amide derivate**

For amine coupling the respective benzoxazole derivative received in first synthesis step (various amounts) was dissolved in DMF and 1.1 mol equiv. respective carboxylic acid was added. While stirring, 1 mol equiv. HATU and 3 mol equiv. DIPEA were added, and the reaction was kept at room temperature until all starting material was converted into the product. The amide was precipitated by adding ice to the reaction. Purification was achieved by column chromatography and the pure product was eluted with constant 3:1 Hexane: Ethyl Acetate solvent mixture.

A01 - *N*-(2-phenylbenzo[*d*]oxazol-5-yl)benzamide

Yield: 57.2 mg (91%); $^1$H-NMR (400 MHz, Methanol-$d_4$) δ 8.25 – 8.20 (m, 2H), 8.19 (d, J = 2.1 Hz, 1H), 7.99 – 7.92 (m, 2H), 7.68 (dd, J = 8.8, 2.0 Hz, 1H), 7.62 (d, J = 8.8 Hz, 1H), 7.60 – 7.47 (m, 6H); $^{13}$C-NMR (151 MHz, Methanol-$d_4$) δ 168.79, 165.18, 148.68, 142.77, 136.93, 135.87, 132.92, 132.74, 130.00, 129.43, 128.45, 128.42, 127.63, 120.57, 113.17, 111.37; ESI-MS *m/z* = 315.2 [M + H$^+$]

A02 - *N*-(2-phenylbenzo[*d*]oxazol-5-yl)propionamide

Yield: 45.7 mg (85%); $^1$H-NMR (400 MHz, Methanol-$d_4$) δ 8.24 – 8.18 (m, 2H), 8.09 (d, J = 2.1 Hz, 1H), 7.62 – 7.52 (m, 4H), 7.50 (dd, J = 8.8, 2.1 Hz, 1H), 2.43 (q, J = 7.6 Hz, 2H), 1.24 (t, J = 7.6 Hz, 3H); $^{13}$C-NMR (151 MHz, Methanol-$d_4$) δ 175.31, 165.24, 148.41, 142.88, 137.29, 133.01, 130.12, 128.48, 127.79, 119.50, 112.03, 111.45, 30.95, 10.24; ESI-MS *m/z* = 267.2 [M + H$^+$]

A03 - *N*-(2-phenylbenzo[*d*]oxazol-5-yl)pentanamide

Yield: 52.8 mg (89%); $^1$H-NMR (400 MHz, Methanol-$d_4$) δ 8.21 (dd, J = 7.9, 1.9 Hz, 2H), 8.09 (d, J = 2.0 Hz, 1H), 7.61 – 7.54 (m, 4H), 7.51 (dd, J = 8.8, 2.1 Hz, 1H), 2.41 (t, J = 7.6 Hz, 2H), 1.71 (tt, J = 7.7, 6.5 Hz, 2H), 1.43 (h, J = 7.4 Hz, 2H), 0.98 (t, J = 7.4 Hz, 3H); $^{13}$C-NMR (151 MHz, Methanol-$d_4$) δ 173.37, 163.93, 147.13, 141.56, 135.90, 131.71, 128.82, 127.21, 126.47, 118.29, 110.83, 110.16, 36.44, 27.71, 22.10, 12.96; ESI-MS *m/z* = 295.2 [M + H$^+$]

A04 - *N*-(2-phenylbenzo[*d*]oxazol-5-yl)isonicotinamide

Yield: 60.8 mg (92%); $^1$H-NMR (400 MHz, Methanol-$d_4$) δ 8.75 (t, J = 3.6 Hz, 2H), 8.23 (dd, J = 8.0, 2.9 Hz, 3H), 7.95 – 7.90 (m, 2H), 7.71 (dt, J = 8.8, 2.0 Hz, 1H), 7.68 – 7.64 (m, 1H), 7.58 (tdd, J = 10.3, 8.6, 5.4, 2.6 Hz, 3H); $^{13}$C-NMR (151 MHz, Methanol-$d_4$) δ 166.06, 165.34, 150.76, 148.93, 144.14, 142.86, 136.43, 132.99, 130.03, 128.47, 127.59, 122.98, 120.39, 113.18, 113.15, 111.54; ESI-MS *m/z* = 316.1 [M + H$^+$]

A05 - *N*-(2-phenylbenzo[*d*]oxazol-5-yl)nicotinamide

Yield: 59.8 mg (91%); $^1$H-NMR (400 MHz, Methanol-$d_4$) δ 9.12 (d, J = 2.2 Hz, 1H), 8.72 (dd, J = 4.9, 1.6 Hz, 1H), 8.37 (dt, J = 7.9, 2.0 Hz, 1H), 8.24 – 8.18 (m, 3H), 7.69 (dd, J = 8.8, 2.1 Hz, 1H), 7.64 (dd, J = 8.8, 1.8 Hz, 1H), 7.62 – 7.53 (m, 4H); $^{13}$C-NMR (151 MHz, Methanol-$d_4$) δ 166.12, 165.28, 152.55, 149.21, 148.83, 142.82, 137.20, 136.58, 132.96, 132.32, 130.01, 128.45, 127.59, 124.94, 120.41, 113.14, 111.50; ESI-MS *m/z* = 316.2 [M + H$^+$]

A06 - *N*-(2-phenylbenzo[*d*]oxazol-5-yl)picolinamide

Yield: 54.8 mg (87%); $^1$H-NMR (400 MHz, Methanol-$d_4$) δ 8.72 (d, J = 4.4 Hz, 1H), 8.37 (d, J = 2.0 Hz, 1H), 8.27 – 8.22 (m, 3H), 8.02 (td, J = 7.7, 1.7 Hz, 1H), 7.75 (dd, J = 8.8, 2.1 Hz, 1H), 7.68 (d, J = 8.7 Hz, 1H), 7.64 – 7.55 (m, 4H); $^{13}$C-NMR (151 MHz, Methanol-$d_4$) δ 165.38, 164.40, 150.83, 149.54, 148.80, 143.01, 138.93, 136.36, 133.02, 130.09, 128.51, 127.89, 127.72, 123.34, 119.92, 112.40, 111.65; ESI-MS *m/z* = 316.2 [M + H$^+$]

A07 - 2-(dimethylamino)-*N*-(2-phenylbenzo[*d*]oxazol-5-yl) acetamide

Yield: 38.5 mg (65%); $^1$H-NMR (400 MHz, Methanol-$d_4$) δ 8.20 (d, J = 6.5 Hz, 2H), 8.12 (d, J = 2.0 Hz, 1H), 7.62 – 7.51 (m, 5H), 3.16 (s, 2H), 2.40 (s, 6H); $^{13}$C-NMR (151 MHz, Methanol-d4) δ 170.88, 165.18, 148.51, 142.79, 136.27, 132.90, 129.97, 128.40, 127.60, 119.48, 112.09, 111.45, 63.96, 45.99; ESI-MS *m/z* = 296.3 [M + H$^+$]

A08 - *N*-(2-phenylbenzo[*d*]oxazol-5-yl)pyrimidine-5-carboxamide

Yield: 55.1 mg (88%); $^1$H-NMR (400 MHz, Methanol-$d_4$) δ 9.32 (s, 1H), 9.31 (s, 2H), 8.24 (d, J = 1.5 Hz, 1H), 8.23 (d, J = 1.4 Hz, 2H), 7.73 – 7.65 (m, 2H), 7.61 – 7.56 (m, 3H); $^{13}$C-NMR (151 MHz, Methanol-$d_4$) δ 165.44, 164.02, 160.96, 157.31, 149.01, 142.95, 136.46, 133.07, 130.19, 130.09, 128.52, 127.65, 120.31, 113.11, 111.64; ESI-MS *m/z* = 317.2 [M + H$^+$]

A09 - *N*-(2-phenylbenzo[*d*]oxazol-5-yl)pyrimidine-4-carboxamide

Yield: 56.3 mg (88%); $^1$H-NMR (400 MHz, DMSO-$d_6$) δ 11.04 (s, 1H), 9.43 (d, J = 1.4 Hz, 1H), 9.13 (d, J = 5.0 Hz, 1H), 8.40 (dd, J = 4.3, 2.1 Hz, 1H), 8.23 – 8.19 (m, 2H), 8.17 (dd, J = 5.1, 1.4 Hz, 1H), 7.93 (ddd, J = 8.8, 3.7, 2.1 Hz, 1H), 7.78 (d, J = 8.8 Hz, 1H), 7.64 – 7.60 (m, 3H); $^{13}$C-NMR (151 MHz, DMSO-$d_6$) δ 163.45, 161.65, 160.11, 158.11, 156.91, 147.43, 141.90, 135.30, 132.27, 129.56, 127.54, 126.63, 119.23, 119.20, 111.85, 110.96; ESI-MS *m/z* = 317.2 [M + H$^+$]

A10 - *N*-(2-phenylbenzo[*d*]oxazol-5-yl)pyrimidine-2-carboxamide

Yield: 46.3 mg (74%); $^1$H-NMR (400 MHz, Methanol-$d_4$) δ 9.01 (d, J = 4.9 Hz, 2H), 8.38 (d, J = 2.1 Hz, 1H), 8.23 (dd, J = 7.7, 1.9 Hz, 2H), 7.78 (dd, J = 8.8, 2.1 Hz, 1H), 7.68 (s, 1H), 7.67 (d, J = 5.0 Hz, 1H), 7.62 – 7.52 (m, 3H); $^{13}$C-NMR (151 MHz, Methanol-$d_4$) δ 164.16, 161.05, 157.68, 157.33, 147.72, 141.73, 134.93, 131.79, 128.84, 127.28, 126.44, 123.11, 118.78, 111.40, 110.42; ESI-MS *m/z* = 317.2 [M + H$^+$]

A11 - 2-methoxy-*N*-(2-phenylbenzo[*d*]oxazol-5-yl)acetamide

Yield: 47.5 mg (85%); $^1$H-NMR (400 MHz, Methanol-$d_4$) δ 8.19 (ddd, $J$ = 6.5, 4.8, 2.0 Hz, 2H), 8.10 (dt, $J$ = 3.5, 2.0 Hz, 1H), 7.61 – 7.51 (m, 5H), 4.06 (s, 2H), 3.51 (s, 3H); $^{13}$C-NMR (151 MHz, Methanol-$d_4$) δ 170.48, 165.24, 148.71, 142.79, 135.87, 132.95, 130.02, 128.46, 127.62, 120.02, 112.73, 111.46, 72.79, 59.70; ESI-MS $m/z$ = 283.2 [M + H$^+$]

**A12 - $N$-(2-phenylbenzo[$d$]oxazol-5-yl)cyclohexanecarboxamide**
Yield: 55.2 mg (88%); $^1$H-NMR (400 MHz, Methanol-$d_4$) δ 8.20 (dd, $J$ = 6.6, 1.9 Hz, 2H), 8.08 (t, $J$ = 2.5 Hz, 1H), 7.60 – 7.52 (m, 4H), 7.51 (dd, $J$ = 8.8, 2.1 Hz, 1H), 2.38 (tdd, $J$ = 11.2, 4.6, 2.2 Hz, 1H), 1.91 (dd, $J$ = 13.7, 2.0 Hz, 2H), 1.84 (dt, $J$ = 12.9, 3.4 Hz, 2H), 1.73 (dt, $J$ = 13.6, 3.5 Hz, 1H), 1.55 (qd, $J$ = 12.6, 3.4 Hz, 2H), 1.37 (q, $J$ = 12.9 Hz, 2H), 1.29 (tt, $J$ = 12.9, 3.4 Hz, 1H); $^{13}$C-NMR (151 MHz, Methanol-$d_4$) δ 177.52, 165.07, 148.25, 142.68, 137.17, 132.85, 129.97, 128.36, 127.63, 119.52, 112.03, 111.28, 46.95, 30.47, 26.64, 26.58; ESI-MS $m/z$ = 321.3 [M + H$^+$]

**A16 - $N$-(2-phenylbenzo[$d$]oxazol-5-yl)pyrimidine-2-carboxamide**
Yield: 42.8 mg (54%); $^1$H-NMR (400 MHz, CDCl$_3$) δ 8.22 (dd, $J$ = 8.8, 5.4 Hz, 2H), 7.93 – 7.86 (m,1H), 7.58 – 7.44 (m, 2H), 7.39 (s, 1H), 7.20 (t, $J$ = 8.6 Hz, 2H), 2.40 (t, $J$ = 7.6 Hz, 2H), 1.75 (p, $J$ = 7.6 Hz, 2H), 1.43 (h, $J$ = 7.4 Hz, 2H), 0.96 (t, $J$ = 7.3 Hz, 3H); $^{13}$C-NMR (151 MHz, CDCl$_3$) δ 171.64, 166.24, 163.15, 147.71, 142.59, 135.11, 130.04, 129.95, 123.50, 118.39, 116.45, 116.23, 111.67, 110.57, 37.66, 27.86, 22.57, 13.98; ESI-MS $m/z$ = 313.1 [M + H$^+$]

**A17 - $N$-(2-(4-chlorophenyl)benzo[$d$]oxazol-5-yl)pentanamide**
Yield: 13.8 mg (21%); $^1$H-NMR (400 MHz, CDCl$_3$) δ 8.15 (d, $J$ = 8.4 Hz, 2H), 7.90 (s, 1H), 7.51 (dd, $J$ = 17.7, 8.3 Hz, 4H), 7.39 (s, 1H), 2.40 (t, $J$ = 7.6 Hz, 2H), 1.74 (p, $J$ = 7.6 Hz, 2H), 1.66 (s, 2H), 1.43 (h, $J$ = 7.3 Hz, 2H), 0.96 (t, $J$ = 7.3 Hz, 3H); $^{13}$C-NMR (151 MHz, CDCl$_3$) δ 171.66, 163.06, 147.69, 142.54, 137.99, 135.18, 129.42 (2 C), 129.00 (2 C), 125.65, 118.61, 111.73, 110.63, 37.65, 27.85, 22.57, 13.98; ESI-MS $m/z$ = 329.1 [M + H$^+$]

**A20 - $N$-(2-(3-fluorophenyl)benzo[$d$]oxazol-5-yl)pentanamide**
Yield: 25.5 mg (37%); $^1$H-NMR (400 MHz, DMSO-$d_6$) δ 10.09 (s, 1H), 8.18 (s, 1H), 8.03 (d, $J$ = 7.7 Hz, 1H), 7.93 (d, $J$ = 9.5 Hz, 1H), 7.72 (d, $J$ = 8.8 Hz, 1H), 7.70 – 7.63 (m, 1H), 7.54 (d, $J$ = 8.9 Hz, 1H), 7.52 – 7.46 (m, 1H), 2.34 (t, $J$ = 7.4 Hz, 2H), 1.60 (p, $J$ = 7.4 Hz, 2H), 1.34 (dq, $J$ = 14.4, 7.3 Hz, 2H), 0.91 (t, $J$ = 7.3 Hz, 3H); $^{13}$C-NMR (151 MHz, DMSO-$d_6$) δ 171.44, 161.16, 146.24, 141.52, 136.86, 131.80, 131.72, 128.66, 123.53, 117.90, 114.00, 113.76, 110.91, 109.88, 36.21, 27.30, 21.92, 13.84; ESI-MS $m/z$ = 313.1 [M + H$^+$]

**A21 - $N$-(2-(2-fluorophenyl)benzo[$d$]oxazol-5-yl)pentanamide**
Yield: 36.9 mg (54%); $^1$H-NMR (400 MHz, DMSO-$d_6$): δ 10.09 (s, 1H), 8.25 – 8.17 (m, 2H), 7.74 (d, $J$ = 8.8 Hz, 1H), 7.72 – 7.65 (m, 1H), 7.58 – 7.52 (m, 1H), 7.51 – 7.41 (m, 2H), 2.34 (t, $J$ = 7.5 Hz, 2H), 1.60 (p, $J$ = 7.5 Hz, 2H), 1.34 (h, $J$ = 7.3 Hz, 2H), 0.91 (t, $J$ = 7.3 Hz, 3H); $^{13}$C-NMR (101 MHz, DMSO-$d_6$): δ 171.43, 146.02, 141.27, 136.80, 134.15 (2 C), 130.52, 125.30, 117.87, 117.42, 117.21, 114.61, 110.84, 109.85, 36.21, 27.31, 21.92, 13.84; ESI-MS $m/z$ = 313.1 [M + H$^+$]. HRMS $m/z$ = 313.1358 (calc 313.1347).

**A22 - $N$-(2-(3-chlorophenyl)benzo[$d$]oxazol-5-yl)pentanamide**
Yield: 34.2 mg (51%); $^1$H-NMR (400 MHz, DMSO-$d_6$) δ 10.09 (s, 1H), 8.20 – 8.10 (m, 3H), 7.74 - 7.68 (m, 2H), 7.64 (t, $J$ = 7.8 Hz, 1H), 7.54 (d, $J$ = 8.8 Hz, 1H), 2.34 (t, $J$ = 7.4 Hz, 2H), 1.60 (p, $J$ = 7.3 Hz, 2H), 1.34 (h, $J$ = 7.3 Hz, 2H), 0.91 (t, $J$ = 7.3 Hz, 3H); $^{13}$C-NMR (101 MHz, DMSO-$d_6$): δ 171.44, 161.51, 146.26, 141.50, 136.86, 134.06, 131.76, 131.45, 128.45, 126.72, 125.91, 117.92, 110.92, 109.88, 36.21, 27.30, 21.92, 13.83; ESI-MS $m/z$ = 329.1 [M + H$^+$]

**A24 - $N$-(2-(2-chlorophenyl)benzo[$d$]oxazol-5-yl)pentanamide**
Yield: 61.3 mg (57%); $^1$H-NMR (400 MHz, DMSO-$d_6$): δ 10.11 (s, 1H), 8.22 (s, 1H), 8.14 (d, $J$ = 7.7 Hz, 1H), 7.77 – 7.69 (m, 2H), 7.64 (t, $J$ = 7.6 Hz, 1H), 7.57 (t, $J$ = 7.8 Hz, 2H), 2.35 (t, $J$ = 7.4 Hz, 2H), 1.61 (p, $J$ = 7.3 Hz, 2H), 1.35 (dq, $J$ = 14.2, 7.1 Hz, 2H), 0.91 (t, $J$ = 7.3 Hz, 3H); $^{13}$C-NMR (101 MHz, DMSO-$d_6$): δ 171.45, 160.83, 146.06, 141.23, 136.77, 132.96, 132.13, 131.96, 131.35, 127.84, 125.53, 118.00, 110.89, 110.01, 36.22, 27.31, 21.92, 13.83; ESI-MS $m/z$ = 329.1 [M + H$^+$]

**A28 - $N$-(2-(pyridin-4-yl)benzo[$d$]oxazol-5-yl)pentanamide**

Yield: 48.3 mg (43%); $^1$H-NMR (400 MHz, DMSO-$d_6$): δ 10.13 (s, 1H), 8.86 – 8.80 (m, 2H), 8.23 (s, 1H), 8.11 – 8.05 (m, 2H), 7.77 (d, $J$ = 8.8 Hz, 1H), 7.58 (dd, $J$ = 8.9, 2.0 Hz, 1H), 2.34 (t, $J$ = 7.5 Hz, 2H), 1.60 (p, $J$ = 7.5 Hz, 2H), 1.34 (h, $J$ = 7.4 Hz, 2H), 0.91 (t, $J$ = 7.3 Hz, 3H); $^{13}$C-NMR (101 MHz, DMSO-$d_6$): δ 171.48, 160.88, 150.94 (2 C), 146.33, 141.37, 137.05, 133.52, 120.81 (2 C), 118.60, 111.17, 110.03, 36.21, 27.28, 21.92, 13.83; ESI-MS $m/z$ = 296.1 [M + H$^+$]

**A29 - $N$-(2-(pyridin-3-yl)benzo[$d$]oxazol-5-yl)pentanamide**
Yield: 57.8 mg (52%); $^1$H-NMR (400 MHz, DMSO-$d_6$) δ 10.10 (s, 1H), 9.36 – 9.32 (m, 1H), 8.80 (q, $J$ = 5.9 Hz, 1H), 8.52 (d, $J$ = 8.1 Hz, 1H), 8.19 (s, 1H), 7.75 (d, $J$ = 8.8 Hz, 1H), 7.65 (dd, $J$ = 7.9, 4.9 Hz, 1H), 7.55 (d, $J$ = 8.9 Hz, 1H), 2.34 (t, $J$ = 7.4 Hz, 2H), 1.60 (p, $J$ = 7.4 Hz, 2H), 1.39 – 1.30 (m, 2H), 0.91 (t, $J$ = 7.3 Hz, 3H); $^{13}$C-NMR (101 MHz, DMSO-$d_6$): δ 171.47, 152.38, 148.02, 146.25, 141.44, 136.89, 134.80, 124.42, 122.90, 117.93, 110.95, 109.87, 36.22, 27.31, 21.93, 13.84; ESI-MS $m/z$ = 296.1 [M + H$^+$]

**A30 - $N$-(2-(pyridin-2-yl)benzo[$d$]oxazol-5-yl)pentanamide**
Yield: 18.7 mg (17%); $^1$H-NMR (400 MHz, DMSO-$d_6$) δ 10.11 (s, 1H), 8.79 (d, $J$ = 4.5 Hz, 1H), 8.32 (d, $J$ = 7.9 Hz, 1H), 8.22 (s, 1H), 8.05 (t, $J$ = 7.8 Hz, 1H), 7.76 (d, $J$ = 8.8 Hz, 1H), 7.65 – 7.60 (m, 1H), 7.56 (d, $J$ = 8.8 Hz, 1H), 2.34 (t, $J$ = 7.4 Hz, 2H), 1.60 (p, $J$ = 7.4 Hz, 2H), 1.34 (dq, $J$ = 14.5, 7.3 Hz, 2H), 0.91 (t, $J$ = 7.3 Hz, 3H); ESI-MS $m/z$ = 296.1 [M + H$^+$]
Amount was not enough for a clear $^{13}$C-NMR spectrum

**A33 - $N$-(2-(thiophen-3-yl)benzo[$d$]oxazol-5-yl)pentanamide**
Yield: quantitative; $^1$H-NMR (400 MHz, DMSO-$d_6$): δ 10.06 (s, 1H), 8.47 (d, $J$ = 4.1 Hz, 1H), 8.11 (d, $J$ = 1.8 Hz, 1H), 7.81 (dd, $J$ = 5.1, 2.9 Hz, 1H), 7.74 (dd, $J$ = 5.1, 1.2 Hz, 1H), 7.67 (d, $J$ = 8.8 Hz, 1H), 7.50 (dd, $J$ = 8.8, 2.0 Hz, 1H), 2.33 (t, $J$ = 7.5 Hz, 2H), 1.60 (dt, $J$ = 15.1, 7.5 Hz, 2H), 1.37 – 1.30 (m, 2H), 0.91 (t, $J$ = 7.3 Hz, 3H); $^{13}$C-NMR (101 MHz, DMSO-$d_6$): δ 171.36, 159.81, 145.77, 141.59, 136.64, 129.45, 128.76, 128.33, 126.31, 117.22, 110.53, 109.71, 36.19, 27.31, 21.92, 13.83; ESI-MS $m/z$ = 301.1 [M + H$^+$]

**A34 - $N$-(2-(thiophen-2-yl)benzo[$d$]oxazol-5-yl)pentanamide**
Yield: quantitative; $^1$H-NMR (400 MHz, DMSO-$d_6$): δ 10.07 (s, 1H), 8.11 – 8.06 (m, 1H), 8.01 – 7.92 (m, 2H), 7.68 (d, $J$ = 8.8 Hz, 1H), 7.52 (dd, $J$ = 8.8, 2.1 Hz, 1H), 7.31 (dd, $J$ = 4.7, 4.0 Hz, 1H), 2.34 (t, $J$ = 7.5 Hz, 2H), 1.60 (dt, $J$ = 15.0, 7.5 Hz, 2H), 1.35 (dt, $J$ = 14.8, 7.4 Hz, 2H), 0.91 (t, $J$ = 7.3 Hz, 3H); $^{13}$C-NMR (101 MHz, DMSO-$d_6$): δ 171.38, 158.99, 145.83, 141.60, 136.80, 131.99, 130.54, 128.99, 128.57, 117.31, 110.54, 109.51, 36.19, 27.30, 21.92, 13.83; ESI-MS $m/z$ = 301.1 [M + H$^+$]

**A39 - $N$-(2-($p$-tolyl)benzo[$d$]oxazol-5-yl)pentanamide**
Yield: 71 mg (57%); $^1$H-NMR (400 MHz, DMSO-$d_6$): δ 10.06 (s, 1H), 8.12 (s, 1H), 8.07 (d, $J$ = 8.0 Hz, 1H), 7.68 (d, $J$ = 8.8 Hz, 1H), 7.50 (d, $J$ = 9.8 Hz, 1H), 7.42 (d, $J$ = 8.1 Hz, 2H), 2.41 (s, 3H), 2.33 (t, $J$ = 7.5 Hz, 2H), 1.60 (p, $J$ = 7.4 Hz, 2H), 1.34 (dq, $J$ = 14.5, 7.3 Hz, 2H), 0.91 (t, $J$ = 7.3 Hz, 3H); $^{13}$C-NMR (101 MHz, DMSO-$d_6$): δ 171.37, 163.09, 146.11, 142.19, 141.74, 136.63, 129.97 (2 C), 127.26 (2 C), 123.76, 117.23, 110.60, 109.70, 36.21, 27.32, 21.93, 21.23, 13.84; ESI-MS $m/z$: 309.2 [M + H$^+$]. HRMS $m/z$ = 309.1599 (calc. 309.1598).

**A40 - $N$-(2-($m$-tolyl)benzo[$d$]oxazol-5-yl)pentanamide**
Yield: quantitative; $^1$H-NMR (400 MHz, DMSO-$d_6$): δ 10.15 (s, 1H), 8.22 (d, $J$ = 1.8 Hz, 1H), 8.10 – 8.03 (m, 2H), 7.77 (d, $J$ = 8.8 Hz, 1H), 7.61 – 7.49 (m, 3H), 2.60 – 2.56 (m, 1H), 2.42 (t, $J$ = 7.5 Hz, 2H), 1.68 (p, $J$ = 7.5 Hz, 2H), 1.42 (h, $J$ = 7.3 Hz, 2H), 0.99 (t, $J$ = 7.3 Hz, 3H); $^{13}$C-NMR (101 MHz, DMSO-$d_6$): δ 171.39, 163.03, 146.17, 141.69, 138.83, 136.67, 132.68, 129.31, 127.67, 126.41, 124.47, 117.42, 110.68, 109.78, 36.21, 27.32, 21.93, 20.94, 13.84; ESI-MS $m/z$: 309.2 [M + H$^+$]. HRMS $m/z$ = 309.1605 (calc. 309.1598).

**A41 - $N$-(2-($o$-tolyl)benzo[$d$]oxazol-5-yl)pentanamide**
Yield: 68 mg (55%); $^1$H-NMR (400 MHz, DMSO-$d_6$): δ 10.07 (s, 1H), 8.18 (s, 1H), 8.11 (d, $J$ = 7.7 Hz, 1H), 7.70 (d, $J$ = 8.8 Hz, 1H), 7.54 – 7.47 (m, 2H), 7.43 (dd, $J$ = 14.8, 7.3 Hz, 2H), 2.74 (s, 3H), 2.34 (t, $J$ = 7.5 Hz, 2H), 1.60 (p, $J$ = 7.4 Hz, 2H), 1.34 (dq, $J$ = 14.3, 7.2 Hz, 2H), 0.91 (t, $J$ = 7.3 Hz, 3H); $^{13}$C-NMR (101 MHz, DMSO-$d_6$): δ 171.39, 163.17, 145.66, 141.67, 138.28, 136.53, 131.98, 131.36, 129.51, 126.50, 125.49, 117.46, 110.60,

109.91, 36.23, 27.34, 21.92 (2 C), 13.84; ESI-MS *m/z*: 309.1 [M + H$^+$]. HRMS m/z = 309.1605 (calc. 309.1598).

**Reporting summary**

Further information on research design is available in the Nature Portfolio Reporting Summary linked to this article.

## Data availability

The data sheet for molecular structure of compounds tested by our lab, as well as where compounds were acquired and their catalog number, are included in the supplemental information as Supplementary Data 1. The data sheet for compound NMR spectra and HRMS data is included in the supplemental information as Supplementary Data 2. Source data are provided with this paper. Original western blots and any additional information required to reanalyze the data reported in this paper is available on the Texas Data Repository https://doi.org/10.18738/T8/JTBZLU.

## Materials availability

Samples of created compounds may be available under an appropriate MTA, pending availability. Synthetic procedures are provided in the method details. All enquiries should be directed to the lead contact.

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

## Acknowledgements

This work was supported by NIH grants NS093382 (J.H.), NS108115 (J.H., S.J.H.), AG053391 (J.H.), HL063762 (J.H.). J.H. was further supported by Blue Field Project to Cure FTD, BrightFocus Foundation (A20135245 and A2016396S), Harrington Discovery Institute, the Alzheimer's Association, the Kleberg Foundation and Circle of Friends Pilot Synergy. We thank the UT Southwestern Proteomics Core especially Andrew Lemmoff for their expertise. We thank Xiaoyu Wang and the resources and expertise provided by the institutionally supported UT Southwestern Preclinical Pharmacology Core, as well as Shuguang Wei and Anwu Zhou at the high throughput core and their supporting funding from the Cancer Center Support Grant (2P30CA142543-11) and UT Southwestern Medical Center. We thank the PhD fellowship from TransMED – Mainz Research School of Translational Biomedicine and research fellowship from Klaus Tschira Foundation Heidelberg. We thank the UCSF Memory and Aging Center for providing HDF cell lines. Figure 5A, B were drawn in part using images from Servier Medical Art. Servier Medical Art by Servier is licensed under a Creative Commons Attribution 4.0 (https://creativecommons.org/licenses/by/4.0/). Base artwork was not modified. We thank Tamara Terrones, Emily Boyle, Isaac Rocha, Alisa Gilloon, Alex Brennan, Angie Asparza, Kyoung Yoo, and Jared Ferrell-Penniman for their support and excellent technical assistance.

## Author contributions

J.H. planned and conceived the project with input by S.J.H., J.H. and S.J.H. obtained funding. J.H. and B.C. designed and initiated the luciferase screen. R.T. and J.H. supervised the execution of the project. R.T., C.G., G.C.W., J.H. and J.R. designed experiments. C.G, J.L, and Y.A. synthesized analogs. R.T., C.G., G.C.W. and D.D. performed experiments, analyzed data and compiled data. B.P. oversaw small molecule HTS. N.W. oversaw PK experiments. J.R. oversaw medicinal chemistry. R.T., C.G. and G.C.W. wrote the manuscript and D.F. and Z.R. edited the manuscript.

## Competing interests

J.H. is a co-founder of Reelin Therapeutics Inc. and coinventor of a patent related to anti-Reelin strategies (Application Number: 15/763,047 and Publication Number: 20180273637). The remaining authors declare no competing interest.
