## [Peer Review File · Nature Communications]

Benzoxazole-derivatives enhance progranulin expression and reverse the aberrant lysosomal proteome caused by GRN haploinsufficiencyREVIEWER COMMENTS

Reviewer #1 (Remarks to the Author):

The authors first screened a 200K library for transcriptional activators of a progranulin promoter-luciferase in N2 cells. Through a series of secondary screens, they narrowed their large hit list and settled upon a promising candidate, C40. Through SAR, they optimized their lead candidate for a number of criteria, including solubility, lipophilicity, and efficacy. They further test the analogs A21 and A41 compounds in a Grn^{+/-} mice after single and multiple treatments, in a time course, for PK and lysosomal improvement. For the latter, this consists of proteomic normalization of isolated lysosomes after compound treatment.

There are many positive aspects to this study, including the impressive initial screening effort, the SAR to improve the lead candidates, the multiple follow-up screens and rigorous use of different cell and mouse models. However, there are issues that limit the overall impact and interpretability of this study.

- 1) The screening effort, while impressive in scope, only will identify transcriptional activators of PGRN, rather than protein stabilizing molecules. This should be acknowledged in the manuscript.
- 2) The Western blots only show a single progranulin band. Often, there are multiple progranulin bands and it is particularly important to know if cleavage fragments are seen with the compounds. Full blots should be included. In addition, the quantification in Figure 3D does not make it clear if the full-length or cleaved progranulin is being measured. Again, the full Western blots should be included.
- 3) Additional characterization of the progranulin produced, including what portion is secreted and/or cleaved, would be useful.
- 4) While the improvement in lysosomal protein profile after A41 treatment is impressive, it is not an actual functional improvement in lysosomal function. The ability of compounds to normalize progranulin functional outcomes would further improve the manuscript.
- 5) On page 11, the conclusion that A41 is “safe” is an overstatement of the experimental results.

Reviewer #2 (Remarks to the Author):

Upon the heterozygous loss-of-function mutations in the GRN gene as a major cause of FTD and potential function of small molecules on regulation of GRN expression reported in literature, the authors here described the finding of a few small molecules by both virtual screening and practical preparation which were used to enhance PGRN levels from the remaining functional GRN allele within the brain, with an attempt to develop small molecule drugs for the treatment of FTD. Compared with the earlier work to boost PGRN levels in FTD models, application of these small molecules could effectively restore PGRN levels back to normal in vitro and in vivo, and protect against the neurodegeneration caused by PGRN loss. They have carefully made the research and provided substantial chemical and biological evidences to support their conclusions. In my opinion, the work is of much significance and well organized, which can be accepted for publication in this journal after addressing the following points.

1. As other researchers have also reported that increasing the expression of progranulin protein levels will be therapeutic to FTD previously, it is unnecessary for the authors to exaggerate their hypothesis that “restoring PGRN levels back to normal will protect against the neurodegeneration caused by PGRN loss”.
2. High resolution ESI-MS data should be provided to ensure the accuracy of all the new compounds. Moreover, purity of representative compounds like compounds C40, C41, C127 and analogs A21, A39, A40 and A41 should be >95% as detected by HPLC.
3. In the first in vitro test, nine compounds were found to be capable of increasing PGRN protein to approximately the target value compared to vehicle levels, however, only three compounds (C40, C41, and C127) enhanced PGRN levels in an in vivo model of FTD with a statistically significant increase. It would better to make SAR analysis on the difference of in vitro and in vivo tests among these nine compounds.
4. In the analog compounds of C40, no alkyl substituents for R2 were used to take the place of a phenyl group, while alkyl substituents for R1 were tried to replace a furanyl moiety. Please give the reason. Besides, did the authors consider the substitution oppositely?
5. They attributed the limited efficacy of C40 and C127 by their ability to dissolve in solution in vivo and in vitro. But I do not think the analog compounds A21 and A41 containing an alkyl chain showing higher activity in contrast to their parent compounds can only be due to the solubility. In fact, as they stated later that a molecule to be able to cross the blood-

brain-barrier (BBB), lipophilicity, MW, flexibility and TPSA are crucial parameters for developing drugs to cure diseases in brain.

6. Compound A03 should be used as positive control in the determination of activity of four analogs A21, A39, A40 and A41 in PGRN protein of U-373 MG glioblastoma cells in Figure 3. I wonder why they treated U-373 MG cells at such a high concentration (100 μ M) of each analog? Besides, in the following HDF cell tests, SAHA should be used as reference.

7. On page 11, the authors summarized that the analogs were more active in vivo compared to the original compounds, possibly due to increased stability and subsequent increased dosing. This is contradictory to what they claim on page 8. Please explain about it.

8. In validating analogs in human HDF cells from FTD patients, HDF cells were treated using A21 and A41 for 24 hours for mRNA and protein analysis. While in measuring the levels of cleaved TPP1, HDF cells from FTD patients were treated using A41 for 72 hours. Please state why big different incubation time was adopted.

9. In the mechanistic study, molecular docking between GRN protein and C40 and its analogs (A21 and A41) are highly recommended to undertake if the GRN in PDB is available. In addition to TFEB activation, further pharmacological mechanism of compound A41 reversing an aberrant lysosomal proteome phenotype in PGRN-deficient models should be explored as PGRN-regulated signaling pathways and gene knockout analysis.

10. The safety profile of A41 needs to be further investigated as histopathological evaluation of brain tissue and several neurotoxicity assays.

11. In Figure S1, please explain why non-small cell lung cancer (NSCLC) cell lines were used to assess the toxicity of the compounds.

Besides, there are some minor errors or shortcomings needing to be corrected.

(i) In Figure 1B, the protein bands of β -actin were inconsistent in C107-treated N2A cells, which could account for some observed changes of experimental condition, please be consistent. Also, the protein bands in Figures 1D and 1F were not clear compared to these in Figure 1B, which need to be explained and rectified.

(ii) On page 10, "Figure S2B" was probably cited in the incorrect place. Page 11, line 3, it was A40, not C40.

Reviewer #3 (Remarks to the Author):

In this manuscript the authors provide a description of a relatively thorough screening paradigm to identify small molecule therapeutic candidates intended to boost PGRN production in patients with GRN haploinsufficiency. Using a N2A cell model with a PGRN dependent luciferase promoter, they screened about 200,000 compounds for candidates that robustly upregulated PGRN, then screened candidate toxicity using NSCLC cell line and ATP-based assay, and selected candidates that boosted brain parenchymal PGRN after intraventricular introduction in a mouse model (though the mouse line is not clearly stated). Next, two candidates (C40 and C127) were assessed using in vitro assessments of liver metabolism, plasma stability, and intestinal permeability, followed by en vivo assessments of BBB permeability after intraperitoneal injections in mice. The brain C_{max} C40 was reassessed after oral gavage (confirming a lower C_{max} and long t_{max}). Finally, the PD effect of intraperitoneal C40 and C127 was assessed in sacrificed Grn^{-/+} mice using western blot assessments of whole brain samples. Of note their initial two primary candidates had limited solubility in solution, so a SAR-mapping approach was then used to optimize for solubility and BBB permeability. Two resulting candidates A21 and A41 upregulated PGRN in their N2A cell model, U-373 Upsala Glioblastoma cells, and finally in fibroblasts reportedly derived from patients with GRN haploinsufficiency. IP delivery of both new candidates reportedly normalized PGRN in Grn^{-/+} mice and toxicity was not detected for A41. In contrast to C40 the candidates did not however partition preferentially to the brain and they had a small C_{max} (albeit a longer t_{max}). Finally, using a tandem mass-spec proteomic approach, the authors identified a proteomic signature in GRN haploinsufficient fibroblasts and mouse cell lines (relative to respective in species controls) and verified the ability of A41 to ameliorate proteins within this signature, particularly proteins relevant to lysosomal function.

Of note, there appears only one compound that underwent testing of oral bioavailability, C40, and this compound was apparently unable to normalize PGRN levels in Grn^{-/+} mice. The other promising compounds do not appear to have oral bioavailability data. This is a worthwhile point because oral bioavailability likely adversely impacted the efficacy of FRM-0334, an HDAC inhibitor that was selected for previous human trial in GRN haploinsufficiency (Ljubenkov et al). Despite optimization, some of the analog compound candidates, like A41, appear to less-readily cross the blood brain barrier, so there are still

ongoing questions about its viability in future development

In the body of the text is sometime ambiguous what patients the fibroblasts were derived from. The authors should always explicitly state if each patient is symptomatic or asymptomatic and explicitly states if they do or do not have haploinsufficiency.

There appear to be multiple editorial comments in the discussion of the history of earlier trials and the about the relative merits of different progranulin-boosting strategies. For instance the author's include the statement "delivery of viruses and proteins to all regions of the human brain presents significant challenges compared to administration of small molecules." Of note, in-human trials of small molecules to date (see Sha et al regarding nimodipine and Ljubenkov et al regarding HDAC inhibitors) have so far failed to show evidence of a pharmacodynamic impact on CSF PGRN.

The authors do not address the necessary next steps before their candidate compounds can move into trials. While 4 interesting candidates have been identified it seems premature to conclude these compounds will "form a basis for the development of a feasible and cost-effective approach to the treatment of FTD caused by GRN haploinsufficiency". These drugs still need better defined oral bioavailability and PD data in other mammalian models, including non-human primates, before moving into human trials.

Reviewer #4 (Remarks to the Author):

GRN mutations are believed to contribute to the development of frontotemporal dementia by causing progranulin haploinsufficiency. Consequently, enhancing progranulin production from the unaffected allele appears to be a plausible approach for treatment.

The researchers employed luciferase-reporter-assays to conduct high-throughput screens, which enabled them to identify several small compounds capable of restoring the expression level of PGRN in both in vitro and in vivo settings. SAR methodologies were additionally employed to enhance the efficacy and certain pharmacological characteristics of these molecules. Furthermore, an investigation was conducted to assess the effectiveness of increasing progranulin (PGRN) levels in rectifying the abnormal lysosomal proteome in cells from patients with GRN-FTD. This was achieved by employing the LysoIP

method in conjunction with lysosomal proteomics.

I support the publication and congratulate them on this work!

Response to reviewers

Reviewer 1:

1) The screening effort, while impressive in scope, only will identify transcriptional activators of PGRN, rather than protein stabilizing molecules. This should be acknowledged in the manuscript. *We thank the reviewer for this comment and have reworded the manuscript accordingly to address this. "...moved forward to develop novel non-toxic and reliable enhancers of PGRN both in vitro and in vivo. We also aimed for transcriptional activators rather than compounds which stabilize PGRN protein to ensure proper PGRN protein function. To accomplish this objective, we..." Our goal was to identify transcriptional activators to take advantage of the residual functional allele and thereby restore normal PGRN mRNA levels without altering protein maturation, transport, etc.*

2) The Western blots only show a single progranulin band. Often, there are multiple progranulin bands and it is particularly important to know if cleavage fragments are seen with the compounds. Full blots should be included. In addition, the quantification in Figure 3D does not make it clear if the full-length or cleaved progranulin is being measured. Again, the full Western blots should be included.

Full length blots of HDF and N2A treated cells were added to Figure 3 and Figure S4 to further clarify questions about changes in full-length vs. cleaved progranulin.

3) Additional characterization of the progranulin produced, including what portion is secreted and/or cleaved, would be useful.

We have included a comparison of secreted versus internalized progranulin with and without our compounds in N2A cells and found that while internalized progranulin is increased, there is no consistent change of progranulin concentrations in the media (Supplemental Figure 5). However, our compounds increased both full length progranulin and cleaved granulins presumed to reside in the lysosome where they are thought to mediate their biological functions. To this extent, there is also evidence that obligate lysosomal progranulin was able to recapitulate the neurotrophic effects of wild type progranulin the absence of extracellular progranulin, suggesting internal, not external progranulin is the primary role of progranulin function.

4) While the improvement in lysosomal protein profile after A41 treatment is impressive, it is not an actual functional improvement in lysosomal function. The ability of compounds to normalize progranulin functional outcomes would further improve the manuscript.

This is a very good point, but unfortunately typical differences are only seen in GRN-/- models where we cannot test our compounds for improvement since our compounds are PGRN transcriptional activators, and no functional allele is present. Heterozygous animal model systems do not present with robust and consistent differences compared to WT with the exception of those used in this manuscript.

5) On page 11, the conclusion that A41 is “safe” is an overstatement of the experimental results.

We updated the manuscript language to avoid overstating the results of our toxicity assessments.

Reviewer 2:

1. As other researchers have also reported that increasing the expression of progranulin protein levels will be therapeutic to FTD previously, it is unnecessary for the authors to exaggerate their hypothesis that “restoring PGRN levels back to normal will protect against the neurodegeneration caused by PGRN loss”. *We appreciate this reviewer’s point and have corrected our manuscript to avoid exaggerating our hypothesis.*

2. High resolution ESI-MS data should be provided to ensure the accuracy of all the new compounds. Moreover, purity of representative compounds like compounds C40, C41, C127 and analogs A21, A39, A40 and A41 should be >95% as detected by HPLC. *We have included these data in the revised manuscript.*

3. In the first in vitro test, nine compounds were found to be capable of increasing PGRN protein to approximately the target value compared to vehicle levels, however, only three compounds (C40, C41, and C127) enhanced PGRN levels in an in vivo model of FTD with a statistically significant increase. It would better to make SAR analysis on the difference of in vitro and in vivo tests among these nine compounds. *The reviewer is referring to our analysis of initial hits from our primary screen. The data described in Figure 1 reflect our efforts to prioritize compounds from the HTS to select ones with in vivo activity for further profiling. These 9 compounds represent 8 different chemical scaffolds (C40 and C127 are analogs). A structural analysis across 8 distinct chemical scaffolds would not be informative, Accordingly, the data suggested that C40 and C127 were suitable for additional study, as described in the remainder of the text.*

4. In the analog compounds of C40, no alkyl substituents for R2 were used to take the place of a phenyl group, while alkyl substituents for R1 were tried to replace a furanyl moiety. Please give the reason. Besides, did the authors consider the substitution oppositely?

A high priority for the program was to replace the furanyl group because furans can be enzymatically oxidized to form reactive, and therefore potentially toxic, metabolites. For this reason, we aggressively modified the amide substructure with aryl and alkyl groups. The optimization process was iterative, and we identified suitable replacements for the furan. Accordingly, we initially limited our changes to R2 to focus on aryl rings because a) the chemistry to provide the benzoxazole is more reliable with R2 = aryl, and b) early analogs showed a robust SAR with substituted (hetero)aryl rings. Once we found compounds with promising properties, we started to evaluate those in preclinical models, while starting the iterative process again to synthesize more different analogs. Ongoing research will continue to explore the R2 group to further improve potency and facilitate target identification.

5. They attributed the limited efficacy of C40 and C127 by their ability to dissolve in solution in vivo and in vitro. But I do not think the analog compounds A21 and A41 containing an alkyl chain showing higher activity in contrast to their parent compounds can only be due to the solubility. In fact, as they stated later that a molecule to be able to cross the blood-brain-barrier (BBB), lipophilicity, MW, flexibility and TPSA are crucial parameters for developing drugs to cure diseases in brain.

We appreciate this comment and have addressed this language in the manuscript text. In brief, the analog compounds A21 and A41 show higher activity at the same concentrations in N2A cells (Figure S4B), so we can see that in this assay C40 and C127's ability to dissolve in solution is not a factor in its increased activity. The same is true of the 30 μ M treatments in the U-373 MG glioblastoma cells. We have edited the text to reflect that since we are unable to treat C40 and C127 at higher concentrations we cannot say for sure if our analogs' increased PGRN induction in the in vivo and in cell culture experiments is due to increased activity, ability to dissolve in solution (and thus an increase in dosing), or a combination of both.

6. Compound A03 should be used as positive control in the determination of activity of four analogs A21, A39, A40 and A41 in PGRN protein of U-373 MG glioblastoma cells in Figure 3. I wonder why they treated U-373 MG cells at such a high concentration (100 μ M) of each analog? Besides, in the following HDF cell tests, SAHA should be used as reference.

Studies were conducted using 4 successive doses from 100 nM, 300 nM, 1 μ M, 3 μ M, 10 μ M, and 30 μ M, so 100 μ M would be the next step in that dosing chain.

Since 30 μ M treatments of C40 and C127 were unable to induce PGRN increases in any human cell lines tested (data not shown) and 100 μ M treatments were not possible because the compound was not form stable solutions in media, evaluating the analogs in the U-373 cells at 30 μ M and 100 μ M would allow us to verify whether our new analogs were more active than our original compounds and if they had increased stability in media which would allow substantial increases treatment concentrations. This reasoning has been added to the manuscript.

We used progranulin enhancing HDAC inhibitors as positive controls throughout the screening process. Our goal was to identify drugs that could restore WT progranulin expression levels (i.e., to increase progranulin levels by approximately 2-fold in heterozygous cells). Therefore, we used changes from baseline expression without known drugs as our reference point. Since SAHA is not a therapeutic candidate, we did not use it as a reference point for further analysis. This study's purpose was not to identify compounds more potent than SAHA but to identify alternative small molecules with sufficient in vitro and in vivo activity to restore normal progranulin levels. This point has been added to the manuscript.

7. On page 11, the authors summarized that the analogs were more active in vivo compared to the original compounds, possibly due to increased stability and subsequent increased dosing. This is contradictory to what they claim on page 8. Please explain about it.

The manuscript has been updated to better explain this: "Since we are unable to administer C40 at higher concentrations, we cannot test if the analogs have an elevated activity level at higher concentrations [in vivo] compared to C40, so the ability of our analogs to fully restore PGRN levels could be due to increased activity, increased stability in solution and subsequent increased dosing, or both."

8. In validating analogs in human HDF cells from FTD patients, HDF cells were treated using A21 and A41 for 24 hours for mRNA and protein analysis. While in measuring the levels of cleaved TPP1, HDF cells from FTD patients were treated using A41 for 72 hours. Please state why big different incubation time was adopted.

To measure downstream effects of progranulin amplification, we first monitored rising progranulin levels over 24 hours, as seen in our time course studies (Figure 4C). After this, our logic was to provide the cells with 48 hours additional dosing so they could reach a new steady state in response to this increase in progranulin levels before determining the new steady state proteomic signature (including TPP1 levels) in our cellular models.

9. In the mechanistic study, molecular docking between GRN protein and C40 and its analogs (A21 and A41) are highly recommended to undertake if the GRN in PDB is available. In addition to TFEB activation, further pharmacological mechanism of compound A41 reversing an aberrant lysosomal proteome phenotype in PGRN-deficient models should be explored as PGRN-regulated signaling pathways and gene knockout analysis.

Compound binding to GRN cannot be ruled out at this point, but we have no reason to believe that our compounds directly interact with GRN since we are looking for compounds that increase GRN transcription rather than stabilizing the protein itself. Additionally, we aim to increase PGRN from its remaining functional allele in heterozygous systems, thus studying knock-out systems should have no effects, but we agree that it would be a great control to pinpoint the effect of our compounds solely on PGRN increase rather than systemic effects independent of PGRN increase.

10. The safety profile of A41 needs to be further investigated as histopathological evaluation of brain tissue and several neurotoxicity assays.

While we performed a basic, prolonged in vivo toxicity assessment of overall external health, weight and CBC profiling, more complex histopathological and neurotoxicity evaluations will be pursued once the compound structure has been further refined and reached an advanced stage of development.

11. In Figure S1, please explain why non-small cell lung cancer (NSCLC) cell lines were used to assess the toxicity of the compounds.

The toxicity screen in the manuscript was performed by the High Throughput Screening Core at UT Southwestern Medical Center. The original goal was to optimize the library

screening assays using NSCLC cell lines, so these cell lines were used for initial toxicity studies. The screening core does not assess compound toxicity in each cell line used in testing in order to conserve key resources.

Besides, there are some minor errors or shortcomings needing to be corrected.

(i) In Figure 1B, the protein bands of β -actin were inconsistent in C107-treated N2A cells, which could account for some observed changes of experimental condition, please be consistent. Also, the protein bands in Figures 1D and 1F were not clear compared to these in Figure 1B, which need to be explained and rectified.

This change in β -actin was C107 specific and consistent between blots. Since this compound was not further pursued due to lack of in vivo activity, the reasons for the dose dependent decrease in cellular β -actin were not explored.

The bands in Figures 1D and 1F are consistent with each other because the samples are from mouse brains. In order to visualize bands from our mouse brain lysates we utilize a different blotting protocol from the N2A lysate protocol (seen in Figure 1B) (see methods for detailed differences). Also, higher concentration of primary antibody are necessary to visualize the target protein in brain lysate, which in turn increases the background of the blots. We therefore would be unable to harmonize the appearance of the blots due to the inherent difference in experimental procedures.

(ii) On page 10, “Figure S2B” was probably cited in the incorrect place.

Page 11, line 3, it was A40, not C40. *We thank this reviewer for pointing out this error. It has been corrected in the manuscript.*

Reviewer 3:

Of note, there appears only one compound that underwent testing of oral bioavailability, C40, and this compound was apparently unable to normalize PGRN levels in Grn-/+ mice. The other promising compounds do not appear to have oral bioavailability data. This is a worthwhile point because oral bioavailability likely adversely impacted the efficacy of FRM-0334, an HDAC inhibitor that was selected for previous human trial in GRN haploinsufficiency (Ljubenkov et al). Despite optimization, some of the analog compound candidates, like A41, appear to less-readily cross the blood brain barrier, so there are still ongoing questions about its viability in future development.

Oral availability, blood-brain barrier permeability, and other relevant pharmacokinetic properties of this compound and its derivatives will certainly have to be continuously assessed and optimized at later stages of development.

In the body of the text is sometime ambiguous what patients the fibroblasts were derived from. The authors should always explicitly state if each patient is symptomatic or asymptomatic and explicitly states if they do or do not have haploinsufficiency.

The genotype of each patient has been clarified in the text and figures, but the de-identified data from the UCSF Memory and Aging Center did not include information about the clinical status or symptoms in control or haploinsufficient patients due to

privacy regulations.

There appear to be multiple editorial comments in the discussion of the history of earlier trials and the about the relative merits of different progranulin-boosting strategies. For instance the author's include the statement "delivery of viruses and proteins to all regions of the human brain presents significant challenges compared to administration of small molecules." Of note, in-human trials of small molecules to date (see Sha et al regarding nimodipine and Ljubenkov et al regarding HDAC inhibitors) have so far failed to show evidence of a pharmacodynamic impact on CSF PGRN.

In terms of HDAC inhibitors, the findings from Ljubenkov et al are consistent with our findings which showed no effects of SAHA on brain progranulin levels (Figure 1E). Of course, CSF progranulin will need to be measured in humans. Progranulin measurement in mouse CSF is experimentally challenging, therefore we analyzed progranulin levels in whole brain lysate.

The authors do not address the necessary next steps before their candidate compounds can move into trials. While 4 interesting candidates have been identified it seems premature to conclude these compounds will "form a basis for the development of a feasible and cost-effective approach to the treatment of FTD caused by GRN haploinsufficiency". These drugs still need better defined oral bioavailability and PD data in other mammalian models, including non-human primates, before moving into human trials. *We appreciate this comment and have addressed this accordingly in the manuscript.*

REVIEWER COMMENTS

Reviewer #1 (Remarks to the Author):

Overall, we feel that the authors have addressed our concerns and further improved the manuscript in response to the other reviewers. One minor point: to avoid confusion between progranulin cleavage fragments and the gene GRN, an alternative to “GRN” should be used when referred to granulins, in particular in figures. Cleavage product, PGRN fragment or even gran would be more clear.

Reviewer #2 (Remarks to the Author):

In the revised manuscript, the authors have given some reasonable answers or explanations to my concerns, but I am still not satisfied with parts of their replies, because it seems that they have ignored a few of my requirements to provide extra data or make further study to support their work. For instance, it is necessary to offer high resolution ESI-MS data of the new compounds and the purity of representative compounds by HPLC, but I cannot find the data in the manuscript. As for point 4, their explanations, though acceptable, should be given in the text. In point 6, I insist that A03 should be used as positive control in the determination of activity in PGRN protein of U-373 MG glioblastoma cells, because A21, A39, A40 and A41 are its analogues. Finally, in the mechanistic study, they do not want to study the interaction of the compounds with GRN “since we are looking for compounds that increase GRN transcription rather than stabilizing the protein itself”. For the reason they claim, I believe they should undertake molecular docking to investigate the possible interactions between GRN protein and C40 and its analogs (A21 and A41) to see whether their claim is true or not.

Overall, I do not think the manuscript can be accepted before the authors address my all concerns.

Response to Reviewers:

Reviewer 1:

Overall, we feel that the authors have addressed our concerns and further improved the manuscript in response to the other reviewers. One minor point: to avoid confusion between progranulin cleavage fragments and the gene GRN, an alternative to "GRN" should be used when referred to granulins, in particular in figures. Cleavage product, PGRN fragment or even gran would be more clear.

Response:

Thank you for raising this point. To remove this ambiguity, processed GRN peptides have been relabeled GRN 2/3 in the figures and are referred to as GRN fragments 2 and 3 in the text. This nomenclature reflects the specificity of the antibody and more precisely indicates what part of the processed PGRN is being detected.

Reviewer 2:

In the revised manuscript, the authors have given some reasonable answers or explanations to my concerns, but I am still not satisfied with parts of their replies, because it seems that they have ignored a few of my requirements to provide extra data or make further study to support their work. For instance, it is necessary to offer high resolution ESI-MS data of the new compounds and the purity of representative compounds by HPLC, but I cannot find the data in the manuscript. As for point 4, their explanations, though acceptable, should be given in the text. In point 6, I insist that A03 should be used as positive control in the determination of activity in PGRN protein of U-373 MG glioblastoma cells, because A21, A39, A40 and A41 are its analogues. Finally, in the mechanistic study, they do not want to study the interaction of the compounds with GRN "since we are looking for compounds that increase GRN transcription rather than stabilizing the protein itself". For the reason they claim, I believe they should undertake molecular docking to investigate the possible interactions between GRN protein and C40 and its analogs (A21 and A41) to see whether their claim is true or not.

Response:

We apologize for inadvertently omitting the HPLC and high resolution mass spectrometry data in our previous submission. They are now included as supplementary data.

Using A03 as a positive control: We had to resynthesize A03, because this was an obsolete compound that was not further pursued for various reasons. We are uncertain why the reviewer thought that using this discarded compound as a positive control was essential, but we have nevertheless repeated the relevant experiments according to these wishes. The essentially identical reproduced results are now shown together with A03 as the positive control. The results naturally do not change the conclusions. Molecular docking studies were performed as described below. However, we are also not certain of the significance of doing this, since our compounds were selected as transcriptional enhancers of GRN expression. There is no basis to assume that these compounds should interact with either PGRN or one of the processed granulins. We have no reason to believe that this is the case. Moreover, no experimentally validated full-length PGRN structure is available. We therefore used two of the available molecular docking programs (Autodock 4 and Neurosnap) to computationally dock our preferred and most active compound A41 to the AlphaFold human PGRN structure. As expected, no specific interaction was identified. For these reasons, these data were not included in the manuscript.

We have further modified the text according to the reviewer's requests. All changes and additions are highlighted in the revised text.

Figure: Docking of A41 (red) to PGRN (grey) AlphaFold Model (AF- P28799-F1) by using **A** AutoDock4 or **B** the online tool Neurosnap resulted in random binding of A41 to PGRN.

All identified compounds in this manuscript were found in a screen for transcriptional regulators of PGRN. Based on our luciferase based assay, where luciferase is coupled to the GRN promotor, we found our compounds to transcriptionally increase progranulin levels. No part of the screen was designed to identify molecules that directly interact with progranulin. We have no evidence that A41 binds directly to progranulin. However, out of deference to the reviewer, we attempted docking to human progranulin.

AutoDock4 (AutoDockTools-1.5.7) was used to dock A41 to AlphaFold model of full-length human progranulin (AF-P28799-F1). A blind docking approach was used where the complete protein was included in the grid box without predefining a binding pocket. Grid-based energy evaluation and all available rotational degrees of freedom for the small molecule were included by AutoDock4 using empirical free energy scoring function and Lamarckian Genetic Algorithm. Additionally, we used an online platform, Neurosnap (<https://neurosnap.ai/>), to cross-validate our findings from AutoDock4. Here, DiffDockL was implemented and used to dock A41 to the AlphaFold model of progranulin.

No preferred binding pocket was found and only unspecific binding occurred all over the structure with both programs, thus this experiment revealed no compelling docked poses for A41 on progranulin. In the absence of experimental or computational data supporting a direct A41-progranulin interaction, we have not revised the text to discuss this potential mechanism.

REVIEWERS' COMMENTS

Reviewer #2 (Remarks to the Author):

In this version of manuscript, the authors have offered sufficient data and extra test results to support their claims, which dispel my doubts. The manuscript can be now accepted for publication as is.